# FROM ONE TO MANY: TRAJECTORY INVARIANT LEARNING FOR MULTIMODAL LARGE LANGUAGE MODEL EDITING

## ABSTRACT

Knowledge editing emerges as a crucial technique for efficiently correcting incorrect or outdated knowledge in large language models (LLMs). Existing editing methods for unimodal LLM rely on a rigid parameter-to-output mapping, which causes causal-underfit and causal-overfit in cascaded reasoning for Multimodal LLM (MLLM). In this paper, we reformulate MLLM editing as an out-of-distribution (OOD) generalization problem, where the goal is to discern semantic shift with factual shift and thus achieve robust editing among diverse cross-modal prompting. The key challenge of this OOD problem lies in identifying invariant causal trajectories that generalize accurately while suppressing spurious correlations. To address it, we propose `ODEdit`, a plug-and-play invariant learning based framework that optimizes the tripartite OOD risk objective to simultaneously enhance editing reliability, locality, and generality. We further introduce an edit trajectory invariant learning method, which integrates a total variation penalty into the risk minimization objective to stabilize edit trajectories against environmental variations. Theoretical analysis and extensive experiments demonstrate the effectiveness of `ODEdit`. Our code is available at https://anonymous.4open.science/r/ODEdit-2756.

## 1 INTRODUCTION

With rapid applications of large language models (LLM) (Liu et al., 2024), ensuring their knowledge correctness and currency in a cost-efficient manner has become a critical concern. *Knowledge editing* (De Cao et al., 2021; Wang et al., 2023; 2024b) is an emerging technique that supports data-efficient modifications on pre-trained models within a specific scope of knowledge. Existing editing methods have two categories, *i.e.,* , i) **parameter-adjusting** (Meng et al., 2022b; Fang et al., 2024; Jiang et al., 2025) directly tune a subset of parameters in the original model, and ii) **model-extending** (Huang et al., 2023; Hartvigsen et al., 2023; Yu et al., 2024) attaches auxiliary components while keeping the backbone parameters intact. A unifying goal is to promote the precision (*Reliability*) and generalization (*Generality*) of LLM perception on the editing knowledge, without compromising irrelevant knowledge outside the editing scope (*Locality*).

Despite these advances, current studies remain largely confined to unimodal LLMs, leaving open their extension to multimodal LLMs (MLLM) (Cheng et al., 2023; Du et al., 2025; Pan et al., 2024; Guo et al., 2025). As Figure 1(a) illustrates, both parameter-adjusting and model-extending methods operationalize editing as a rigid mapping from parameter modifications $\Delta W$ or auxiliary component modifications $\Delta M$ to output variations $\Delta Y$, distilled from a limited training cases. However, in the Structural Causal Model (SCM) view (Li et al., 2024c; Zhou et al., 2024), the forward computation graph of an MLLM is a structural causal model: each module implements a structural equation, forming a directed causal chain as *unimodal perception → cross-modal alignment → shared semantic reasoning*. Under this structure, any local change to a module or parameter inevitably propagates downstream and alters subsequent states, and their effect are mediated by all later causal mechanisms. Consequently, rigid mapping from any single structure to the edited output in MLLM editing easily induces two issues:

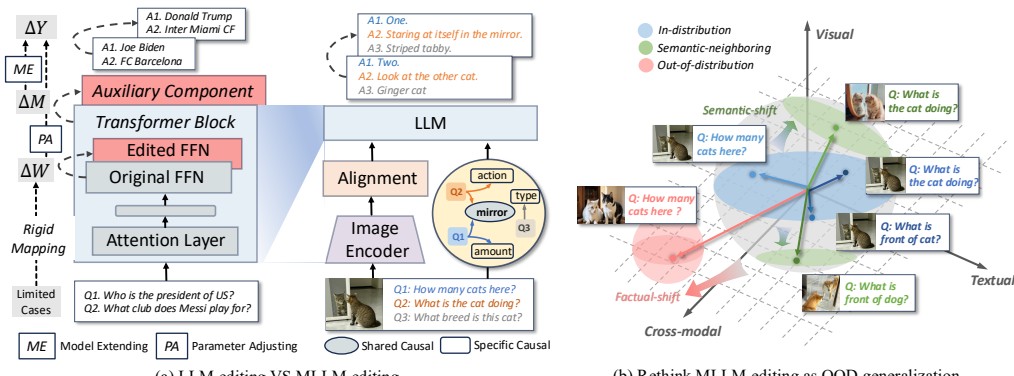

Figure 1: The motivation of `ODEdit`. The left presents why previous editing work targeted at unimodal LLM is not effective in MLLM. The right denotes two shifts in this editing OOD problem.

- **Causal Underfit.** It fails to disentangle the coherent *shared* causal structures that span diverse cross-modal contexts. This precludes MLLM from discovering reusable semantic substrates, reducing editing to a brittle case-wise alignment exercise rather than principled semantic *generality*.

- **Causal Overfit.** Such a mapping drives MLLM to memorize entrenched linkages between recurrent local features and outputs. The inflexible causal chains misguide the model when faced with queries that share local features yet differ in specific causal semantics, thus compromising *locality*.

These limitations raise a critical question: ***How can knowledge editing orchestrate MLLM comprehension adaptively on cross-modal prompting, while balancing semantic transferability and robustness to spurious correlations?***

To this end, we rethink MLLM editing as an out-of-distribution (OOD) cross-modal semantic generalization problem. OOD (Ye et al., 2021; Montasser et al., 2024) originally refers to identifying invariant versus spurious features that drive distribution shifts, enabling the model to generalize to unseen domains. In Figure 1(b), editing MLLM involves partitioning causal scopes in prompts, which has two distributional shifts: **(i) *Semantic-shift* indicates the shift from in-distribution editing scopes to neighboring regions, constituting the intended generalized targets. It refers to meaning-preserving variations that keep the atomic factual content and output-relevant conceptual factors unchanged.** Once the editing process instills the *mirror reflection* principle into MLLM, the model can generalize from a narrowly edited instance (*a tabby cat staring itself in the mirror*) to similar scenarios (*an orange cat or dog doing the same thing*). **(ii) *Factual-shift* denotes the transition from in-distribution to out-of-distribution regions, encompassing extraneous concepts beyond editing scopes. It refers to variations that alter the underlying atomic factual content and modify the model's reasoning-relevant conceptual representation.** The mirror reflection principle should not be overapplied to counting prompts lacking mirror-specific visual features. Building on two shifts, robust MLLM editing requires identifying **invariant trajectories** for cross-modal predictions and removing **spurious factors** that disrupt causal associations.

In this paper, we propose a plug-and-play editing OOD optimization framework for multimodal LLM, termed `ODEdit`, which leverages cross-modal causal trajectory invariant learning to ensure knowledge editing robustness across diverse distributions. To explicitly enhance the MLLM's *discriminative awareness of semantic-shift and factual-shift*, we first introduce a tripartite OOD risk formulation that imposes tailored constraints on in-distribution, semantic-neighboring, and out-of-distribution features. We apply the Kullback-Leibler divergence regularization to preserve locality while developing a maximum mean discrepancy-based metric learning to align representations of edited concepts and their semantic variants. To *discern and stabilize the edit trajectories* across heterogeneous cross-modal environments, we further propose Edit Trajectory Invariant Learning (ETIL). ETIL first reforms the editing OOD objective into an equivalent invariant risk minimization problem, where an environment-aware classifier is introduced to exploit feature invariance and irrelevance. Then, to suppress the sensitivity of the edit trajectory to spurious environmental changes, ETIL integrates a Total Variation factor as the penalty term in the risk estimation. The invariant risk minimization is achieved through a primal-dual optimization strategy, ensuring that the edited model captures reusable causal structures while filtering out superficial correlations.

Main contributions are (1) We revisit the knowledge editing on MLLM from the OOD generalization perspective, and propose a plug-and-play optimization paradigm. (2) We introduce a tripartite OOD risk that imposes tailored constraints on semantic- and factual-shift, and develop a trajectory invariant learning to minimize composed editing risk across diverse cross-modal prompting. (3) We provide theoretical analyses and extensive experiments to validate effectiveness of `ODEdit`.

## 2 PRELIMINARY

**Out-of-Distribution Generalization.** Considering datasets $\mathcal{D}_e := \{x_i^e, y_i^e\}_{i=1}^{n_e}$ collected from diverse training environments $e \in \mathcal{E}_{train}$, the environments correspond to identical random variables assessed under distinct conditions. The dataset $\mathcal{D}_e$ consists of i.i.d. samples drawn from the probability distribution $P(X^e, Y^e)$. OOD generalization targets at learning a predictor $f : \mathcal{X} \rightarrow \mathcal{Y}$ that minimizes the worst-case risk over a broad, potentially unseen set of environments $\mathcal{E}_{\text{all}} \supseteq \mathcal{E}_{\text{train}}$:

$$\mathcal{R}_{\text{OOD}}(f) = \max_{e \in \mathcal{E}_{\text{all}}} \mathbb{E}_{(X^e, Y^e) \sim P^e}[\ell(f(X^e), Y^e)].$$

Here, $\mathbb{E}_{(X^e, Y^e) \sim P^e}[\ell(f(X^e), Y^e)]$ denotes the risk under specific environment $e$, and $\ell$ is a suitable loss function. The set $\mathcal{E}_{\text{all}}$ includes environments not encountered during training.

**Invariant Risk Minimization (IRM).** IRM (Arjovsky et al., 2019; Tan et al., 2023) generalizes invariant features to different environments. Given training data as $\mathcal{D} := \{(x_i, y_i) \in \mathcal{X} \times \mathcal{Y}\}$ where $\mathcal{X}$ and $\mathcal{Y}$ denotes the input and output space. IRM constructs the learning model $\mathcal{X} \rightarrow \mathcal{Y}$ into two parts, *i.e.,* the feature extractor $\Psi : \mathcal{X} \rightarrow \mathcal{H}$ mapping input into the invariant feature space and the classifier $\omega : \mathcal{H} \rightarrow \mathcal{Y}$ predicting based on these features. The empirical risk under environment $e$ is:

$$\mathcal{R}(\omega \circ \Psi, e) = \frac{1}{n} \sum_{i=1}^{n} \mathcal{L}(\omega \circ \Psi(x_i), y_i, e),$$

where $\mathcal{L}$ is the loss function. The original IRM formulation is a bi-level optimization problem:

$$\min_{\omega, \Psi} \sum_{e \in \mathcal{E}_{tr}} \mathcal{R}(\omega \circ \Psi, e) \qquad \text{s.t. } \omega \in \arg\min_{\tilde{w}} \mathcal{R}(\tilde{\omega} \circ \Psi, e), \quad \forall e \in \mathcal{E}.$$

This constraint requires $\omega$ to be optimal for each environment given $\Psi$, encouraging $\Psi$ to extract invariant features. Further, IRMv1 (Arjovsky et al., 2019) provides a surrogate form, which fixes the classifier $\omega$ to a constant scalar and replaces the constraint with a gradient norm penalty:

$$\min_{\Psi} \sum_{e \in \mathcal{E}} \left\{ \mathcal{R}(1 \circ \Psi, e) + \lambda \left\| \nabla_{\omega|_{\omega=1}} \mathcal{R}(\omega \circ \Psi, e) \right\|_2^2 \right\}.$$

## 3 METHODOLOGY

### 3.1 PROBLEM SETTING

**MLLM Editing as OOD Problem.** First, we formulate the knowledge editing task in a MLLM with the out-of-distribution generalization form. Considering the MLLM as a function $\mathcal{M} : \mathcal{I} \times \mathcal{X} \rightarrow \mathcal{Y}$ with parameters $\phi$, which takes the cross-modal prompt $(i_e, x_e)$ consisting of an image $i_e$ and a textual description $x_e$ as input, and generates $y_o$ as the original output. Denote the editing dataset containing facts to be updated as $\mathcal{D}_{\text{edit}}$, we define an environment factor $e \in \mathcal{E}$ which parameterizes the data distribution $\mathcal{P}_e(I, X, Y)$, indicating all the possible causal associations that can occur in testing prompts. The objective of MLLM editing is to update $\phi \rightarrow \phi_e$ for the worst-case risk $\mathcal{R}_{\text{edit}}(\phi_e, e)$ across all conceivable environments:

$$\min_{\phi_e} \max_{e \in \mathcal{E}} \mathbb{E}_{(i_e, x_e, y_e) \sim \mathcal{P}_e(I, X, Y)} \mathcal{R}_{\text{edit}}(\phi_e, (i_e, x_e, y_e), e), \quad (1)$$

Empirically, we assume the testing prompts $\mathcal{D}_{\text{test}}$ are composed of in-distribution data $\mathcal{D}_{\text{in}}$, semantic-neighboring data $\mathcal{D}_{\text{se}}$, and out-of-distribution data $\mathcal{D}_{\text{out}}$. The overall risk is defined as $\mathcal{R}_{\text{edit}} = \mathcal{R}_{\text{rel}} + \mathcal{R}_{\text{gen}} + \mathcal{R}_{\text{loc}}$, demonstrating the composite measure of three editing metrics (Cheng et al., 2023). $\mathcal{R}_{\text{rel}}$,

$\mathcal{R}_{\text{gen}}$, and $\mathcal{R}_{\text{loc}}$ respectively justify the editing performance on three aspects, *i.e.,* editing accuracy on $\mathcal{D}_{\text{IN}}$, generalization ability on $\mathcal{D}_{\text{SE}}$, and side effects on $\mathcal{D}_{\text{OUT}}$, as follows:

$$\mathcal{R}_{\text{rel}} := \mathbb{E}_{(i_e,x_e,y_e)\sim\mathcal{P}_{\mathcal{D}_{\text{IN}}}} \left[ \mathbb{1}\{\mathcal{M}(i_e, x_e; \phi_e(i_e, x_e, y_e)), y_e)\} \right]$$

$$\mathcal{R}_{\text{loc}} := \mathbb{E}_{(i_t,x_t)\sim\mathcal{P}_{\mathcal{D}_{\text{OUT}}}} \left[ \mathbb{1}\{\mathcal{M}(i_t, x_t; \phi_e(i_e, x_e, y_e)) = \mathcal{M}(i_t, x_t; \phi)\} \right] \quad (2)$$

$$\mathcal{R}_{\text{gen}} := \mathbb{E}_{(i_r,x_r)\sim\mathcal{P}_{\mathcal{D}_{\text{SE}}}} \left[ \mathbb{1}\{\mathcal{M}(i_e, x_e; \phi_e(i_e, x_e, y_e)) = \mathcal{M}(i_r, x_r; \phi_e(i_e, x_e, y_e))\} \right]$$

## 3.2 Semantic-Factual Shift Disentanglement

To facilitate MLLM discriminate editing environments between semantic-shift and factual-shift, we first design independent editing risks to evaluate transferability on invariant trajectories and capability to eliminate spurious factors. Our framework aims to construct a unified optimization paradigm that is agnostic to specific editing methods, so it can be incorporated into any parameter-adjusting or model-extending editing approach based on fine-tuning. With the pre-trained multimodal LLM $\mathcal{M}_\phi$ and editing dataset as $\mathcal{D}_{edit}$, we denote the editing model as $f_\theta$. Editing is cast as learning a mapping $\Gamma$ that adapts the model and its parameters guided by the edit instance and $f_\theta$:

$$\mathcal{M}(\phi_e, \theta_e) = \Gamma\left(\mathcal{M}_\phi, f_\theta; (i_e, x_e, y_e)\right)(\cdot), \quad (i_e, x_e, y_e) \in \mathcal{D}_{\text{edit}} \quad (3)$$

Then, to optimize the three objectives outlined in Section 3.1, *i.e.,* reliability, locality, and generality, we propose corresponding risk metrics that are seamlessly integrated into these base editing models. **Reliability Risk.** To ensure precise assimilation of the edited knowledge, we minimize the negative log-likelihood of the target output conditioned on the edit instance:

$$\mathcal{R}_{\text{rel}} = -\log p_{\phi_e}(y_e \mid i_e, x_e), \quad (i_e, x_e, y_e) \in \mathcal{D}_{\text{IN}}, \quad (4)$$

which explicitly maximizes the probability of the desired output $y_e$ for the edited input $(i_e, x_e)$, ensuring accurate cognition on the in-distribution cross-modal semantics.

**Locality Risk.** In order to avoid the edited concepts affecting the interpretation of unrelated content falling within the factual-shift scope, we regularize the editing process by imposing a Kullback–Leibler divergence (Attias, 1999) penalty between the pre- and post-edit output distributions:

$$\mathcal{R}_{\text{loc}} = \text{KL}\left(p_{\phi_e}(\cdot \mid i_{\text{t}}, x_{\text{t}}) \,\|\, p_\phi(\cdot \mid i_{\text{t}}, x_{\text{t}})\right), \quad (i_t, x_t, y_t) \in \mathcal{D}_{\text{OUT}}. \quad (5)$$

This constraint strengthens model capacity to preserve knowledge beyond the designated editing scope, maintaining editorial locality and minimizing unintended side effects.

**Generality Risk.** Previous methods (Mitchell et al., 2022; Zeng et al., 2024) mostly emphasize supervised partitioning of in-scope and out-of-scope knowledge regions, but fall short in achieving semantic generalization, and thus cause issues of causal-underfit or causal-overfit. Thus, we propose a generality risk for extracting invariant trajectories hidden underneath semantic-shift cross-modal prompting. **For each edited instance** $(i_e, x_e)$**, we utilize its rephrase counterparts** $(i_r, x_r)$ **from the benchmark training datasets.** Let $z_{\phi_e}(i, x)$ denote the last hidden states of edited model $\mathcal{M}_{\phi_e}$ for prompt $(i, x)$, we retrieve the distributions of edited prompts and rephrase prompts as $\boldsymbol{Z}_E$ and $\boldsymbol{Z}_R$ respectively. Then we develop a Maximum Mean Discrepancy (MMD) (Tolstikhin et al., 2016) based metric learning to measure the discrepancy between in- and semantic-neighboring distributions. Given the Kernel Hilbert Space $\mathcal{H}$ associated with the Borel measurable kernel $k$, the kernel mean embedding $\boldsymbol{\mu}_{\boldsymbol{Z}_E}$ and $\boldsymbol{\mu}_{\boldsymbol{Z}_R}$ is formulated with the reproducing property as:

$$\boldsymbol{\mu}_{\boldsymbol{Z}_E} = \int_{\mathbb{S}} k(s, \cdot) \boldsymbol{Z}_E(ds) \in \mathcal{H}, \quad \boldsymbol{\mu}_{\boldsymbol{Z}_R} = \int_{\mathbb{V}} k(v, \cdot) \boldsymbol{Z}_R(dv) \in \mathcal{H}, \quad (6)$$

where $s$ and $v$ are random variables with distribution $\boldsymbol{Z}_E$ and $\boldsymbol{Z}_R$. It satisfies the distribution probability density equation that for all functions $f \in \mathcal{F}$:

$$\mathbb{E}\left[f(S)\right] = \langle f, \boldsymbol{\mu}_{\boldsymbol{Z}_E}\rangle_{\mathcal{H}}, \quad \mathbb{E}\left[f(V)\right] = \langle f, \boldsymbol{\mu}_{\boldsymbol{Z}_R}\rangle_{\mathcal{H}}. \quad (7)$$

We deploy the multi-scale Gaussain kernel function $k(x_i, x_j) = \sum_{q=1}^{k} \exp\left(-\frac{\|x_i - x_j\|_2^2}{2\sigma_q^2}\right)$ in $\mathcal{H}$ to simultaneously capture local and global similarity between two instances, where $\sigma_q$ denotes the bandwidth of $q$-th kernel. Based on this, the generality risk in the MMD form is defined as:

$$\mathcal{R}_{\text{gen}} = \mathbb{E}_{z_e, z_e' \sim \boldsymbol{Z}_E}\left[k(\boldsymbol{z}, \boldsymbol{z}_e')\right] + \mathbb{E}_{z_r, z_r' \sim \boldsymbol{Z}_R}\left[k(\boldsymbol{z}, \boldsymbol{z}_r')\right] - 2\mathbb{E}_{z_e \sim \boldsymbol{Z}_E, z_r \sim \boldsymbol{Z}_R}\left[k(\boldsymbol{z}_e, \boldsymbol{z}_r)\right]. \quad (8)$$

### 3.3 Edit Trajectory Invariant Learning

With the editing OOD formulation in Section 3.1 and the overall risk composed of supervised signals on two distributional shifts in Section 3.2, we now introduce an invariant learning paradigm to discern and stabilize the edit trajectories across diverse cross-modal environments. Our goal is optimizing the edited model parameters $\phi_e$ to minimize the risk over all environments, which requires exploiting invariance and specificity in the causal pathways activated by edit.

**Transformation into IRM Problem.** To extract invariant trajectories, we invoke the IRM principle and employ a classifier $\omega$ which maps environment features to predictions (Lai & Wang, 2024).

**Proposition 1** (Equivalence between OOD-$\omega$ and IRM). *Under the condition that the environment variability is channeled through the classifier $\omega$, it satisfies the identity $\mathcal{R}_{edit}(\phi_e, e) \equiv \mathcal{R}_{edit}(\omega(e) \circ \phi_e)$. The OOD editing objective in Eq.(1) admits the following equivalent IRM formulation:*

$$\min_{\phi_e} \max_{\omega \in \Sigma} \mathbb{E}_{(i_e, x_e, y_e) \sim \mathcal{P}_e(I, X, Y)} \mathcal{R}_{edit}(\omega(i_e, x_e, y_e) \circ \phi_e).$$

*Proof.* The proof can be found in Appendix A.1. □

**Invariant Learning in Editing Trajectory.** Directly optimizing the OOD-$\omega$ objective is intractable due to the need to evaluate the supremum over $\mathcal{E}$. To this end, we reformulate it within a measure-theoretic framework inspired by the connection between IRM and Total Variations (TV) (Chan et al., 2006). The TV operator typically employed to measure the global variability bound of a function. For a function $f$ defined on a measure space $(\Omega, \mathcal{F}_\Omega, \nu)$, the TV seminorm is given by

$$TV(f) := \sup \left\{ \int_\Omega f(\nu) \operatorname{div} g(\nu) d\nu : g \in C_c^1(\Omega, \mathbb{R}^d), \|g\|_\infty \leq 1 \right\}, \tag{9}$$

where $g$ is a differentiable vector function supported compactly in $\Omega$ and $\operatorname{div} g$ denotes its divergence. Based on the Coarea Formula (Chan et al., 2006), the canonical TV-$\ell_1$ (Rudin et al., 1992) can be derived to recover a clean signal $f$ from a noisy observation $\tilde{f}$ by solving the variational problem:

$$\inf_{f \in L^2(\Omega)} \left\{ \int_\Omega |\nabla f| + \lambda \int_\Omega (f - \tilde{f})^2 d\nu \right\} \tag{10}$$

Here, TV-$\ell_1$ model pres sharp discontinuities while effectively removing noise and fine-scale details. Correspondingly, we treat the environment-induced variations in the risk function $\mathcal{R}_{\text{edit}}(\omega \circ \phi_e)$ as noise perturbing the ideal and invariant edit trajectory. The goal of editing is to *denoise* the risk, recovering a piecewise-constant profile that is robust to spurious cross-modal prompting changes. Inspired by Lai & Wang (2024), we further absorb TV-$\ell_1$ penalty into our editing IRM objective as

$$\min_{\phi_e} \left\{ \mathbb{E}_\omega[\mathcal{R}_{\text{rel}}(\omega \circ \phi_e) + \mathcal{R}_{\text{loc}}(\omega \circ \phi_e) + \mathcal{R}_{\text{gen}}(\omega \circ \phi_e)] + \lambda_{\phi_e} \left( \mathbb{E}_\omega[|\nabla_\omega \mathcal{R}_{\text{edit}}(w \circ \phi_e)|] \right)^2 \right\}. \tag{11}$$

The first term represents the basic risk of editing, while the second term promotes invariance by encouraging the generalization risk to be insensitive to environmental changes. This form directly addresses the dual requirements of precise *knowledge assimilation* and *controlled generalization*.

**Proposition 2** (IRM-TV objective Achieves Editing OOD with a varying $\lambda$). *The balancing parameter $\lambda$ should vary with editing parameters $\phi_e$ to achieve editing OOD. For each $\phi_e$, if $\mathbb{E}_\omega[|\nabla_\omega \mathcal{R}_{edit}(\omega \circ \phi_e)|] > 0$, there exists a non-negative $\lambda_{\phi_e}$, such that*

$$\max_{e \in \mathcal{E}} \mathcal{R}_{edit}(\phi_e, e) = \mathbb{E}_\omega[\mathcal{R}_{rel}(\omega \circ \phi_e) + \mathcal{R}_{loc}(\omega \circ \phi_e) + \mathcal{R}_{gen}(\omega \circ \phi_e)] + \lambda_{\phi_e} \left( \mathbb{E}_\omega[|\nabla_\omega \mathcal{R}_{edit}(\omega \circ \phi_e)|] \right)^2.$$

*Besides, the optimality of $\phi_e$ for IRM-TV form is equivalent to its optimality for OOD-$\omega$.*

*Proof.* The proof can be found in Appendix A.2. □

**Optimization on Editing IRM-TV.** To solve Eq.(11), we treat $\lambda_{\phi_e}$ as a Lagrangian multiplier and parameterize it as a function $\lambda(\pi, \phi_e)$ of both the editing model parameters $\phi_e$ and an auxiliary dual parameter set $\delta$. We derive the Lagrangian function for the editing IRM-TV objective as

$$\mathcal{G}(\delta, \phi_e) = \mathbb{E}_\omega[\mathcal{R}_{\text{rel}}(\omega \circ \phi_e) + \mathcal{R}_{\text{loc}}(\omega \circ \phi_e) + \mathcal{R}_{\text{gen}}(\omega \circ \phi_e)]] + \lambda(\delta, \phi_e) \left( \mathbb{E}_\omega[|\nabla_\omega \mathcal{R}_{\text{edit}}(\omega \circ \phi_e)|] \right)^2. \tag{12}$$

Denote the risk sum as $\mathcal{R}_{edit}$, we derive it into a primal-dual optimization as (Wang et al., 2025)

$$\min_{\phi_e} \max_{\delta} \mathcal{G}(\delta, \phi_e) := \min_{\phi_e} \left\{ \mathbb{E}_\omega [\mathcal{R}_{\text{edit}}(w \circ \phi_e)] + \max_{\delta} \left[ \lambda(\delta, \phi_e) \left( \mathbb{E}_\omega \left[ \|\nabla_\omega \mathcal{R}_{\text{edit}}(\omega \circ \phi_e)\| \right] \right)^2 \right] \right\},$$

(13)

where the *primal variable* $\phi_e$ is optimized to minimize the overall risk, and *dual variable* $\delta$ is optimized to maximize the TV penalty. To solve it, an adversarial learning procedure is adopted, alternating between updating $\phi_e$ and $\delta$ with adaptive learning rates $\gamma_1$ and $\gamma_2$:

$$\phi_e^{(k+1)} = \phi_e^{(k)} - \gamma_1^{(k)} \cdot \partial_{\phi_e} \mathcal{G}(\delta^{(k)}, \phi_e^{(k)}), \; \delta^{(k+1)} = \delta^{(k)} + \gamma_2^{(k)} \cdot \nabla_\delta \mathcal{G}(\delta^{(k)}, \phi_e^{(k+1)}).$$

(14)

The computation process of gradient $\nabla_\delta \mathcal{G}$ and subgradient $\partial_{\phi_e} \mathcal{G}$ are presented in Appendix A.3. Consequently, after optimizing two variables, we obtain an edit model $\phi_e$ that is both accurate and contained while being robustly generalizable through invariant mechanisms.

## 4 EXPERIMENT

### 4.1 EXPERIMENTAL SETUP

**Datasets & Backbones & Evaluation Metrics.** In line with previous work (Pan et al., 2024), we conduct experiments on the MMEdit benchmark (Cheng et al., 2023), encompassing two sub-tasks, *i.e.,* Editing VQA (E-VQA) and Editing Image Captioning (E-IC). Under this benchmark, we choose BLIP2-OPT (Li et al., 2023) and MiniGPT-4 (Zhu et al., 2023) as the base MLLM. We utilize Reliability, Generality, and Locality (T-Locality and M-Locality) as the evaluation metrics.

**Baseline Methods.** We describe four types of baselines and how we incorporate ODEdit into each method as a plug-and-play optimization framework in Appendix D.3.

**Implementation Details.** We present all implementation details in Appendix D.4.

### 4.2 PERFORMANCE ON ONE-STEP KNOWLEDGE EDITING

To evaluate editing performance, we conduct one-step editing experiments. From Table 1, we can find: 1) **Previous methods fail to achieve balanced performance across all metrics when applied to multimodal editing tasks.** Model-extending methods frequently suffer from poor locality, while parameter-adjusting methods often exhibit limited generality. For instance, SERAC achieves high reliability (97.60) and generality (97.30) with BLIP2 on E-VQA, but its M-Locality drops drastically to 3.21. MEND shows a significant generality gap with the other SOTAs like SERAC. 2) **ODEdit demonstrates strong adaptability across diverse baselines and consistently improves four evaluation metrics.** On E-VQA with MiniGPT-4, T-Patcher+ODEdit improves generality with the promotion ratio as 4.82%. WISE+ODEdit improves M-Locality by 19.2% with MiniGPT-4 on E-VQA, while T-Locality by 17.2% with BLIP2 on E-IC. UniKE+ODEdit outperforms UniKE on all metrics. (3) The balanced improvement across metrics underscores that effective OOD generalization equates to holistic performance elevation, not merely gains in the generality dimension. ODEdit accurately determines the generalization boundary, and its core contribution is extracting invariant editing trajectories to both mitigate causal underfit and causal overfit, thus resolving the trade-off between locality and generality.

### 4.3 PERFORMANCE ON LONG-TERM KNOWLEDGE EDITING

Following Pan et al. (2024), we typically set the $T$-step sequential editing scenario, where the model is edited sequentially for each instance in the editing set with a capacity of $T$. After the $T$-th edit, we evaluate the post-edit MLLM. We report the results for $T = 5$ and $T = 10$ on both E-VQA and E-IC tasks. From Table 2, we find: 1) Unimodal editors like WISE fail catastrophically in multimodal long-term editing, particularly in preserving locality and generality. On E-VQA, WISE's T-Loc. collapses to near zero, demonstrating the rigid editing mapping cannot adaptively modify MLLM's causal reasoning. 2) Even specialized multimodal editors like UniKE exhibit performance decay over time. This indicates that without explicit invariance learning, sequential edits cause interference and erode previously learned knowledge. 3)

Table 1: Overall editing performance (%). *Rel.*, *Gen.*, *T-Loc.*, *M-Loc.*, denote Reliability, Generality, Text Locality, and Image Locality respectively. The higher scores within the same editing backbone are highlighted in bold. All improvements are significant with $p$-value $< 0.05$ based on $t$-tests.

| Model | Method | Editing VQA (E-VQA) | | | | Editing Image Caption (E-IC) | | | |
|---|---|---|---|---|---|---|---|---|---|
| | | Rel.↑ | Gen.↑ | T-Loc.↑ | M-Loc.↑ | Rel.↑ | Gen.↑ | T-Loc.↑ | M-Loc.↑ |
| | Pre-edited | 25.85 | 26.37 | 99.38 | 92.83 | 0 | 0 | 99.79 | 94.93 |
| BLIP2-OPT 2.7B | FT | 100 | 100 | 93.94 | 64.79 | 100 | 0 | 78.79 | 29.58 |
| | IKE | 99.71 | 99.62 | 47.74 | 2.53 | 94.40 | 88.00 | 50.43 | 2.87 |
| | SERAC | 97.60 | 97.30 | 100 | 3.21 | 99.71 | 99.71 | 100 | 2.64 |
| | WISE | 100 | 83.33 | 40.94 | 16.89 | 100 | 85.93 | 33.61 | 11.89 |
| | WISE+ODEdit | **100** | **83.33** | **41.24** | 13.83 | **100** | **87.33** | **39.37** | **14.77** |
| | MEND | 97.80 | 97.20 | 99.68 | 94.23 | 77.90 | 62.80 | 98.14 | 78.86 |
| | MEND+ODEdit | 97.60 | **97.20** | 99.52 | 91.75 | **79.40** | **64.40** | **99.01** | **86.14** |
| | T-Patcher | 80.35 | 77.82 | 87.14 | 85.28 | 72.78 | 72.75 | 71.59 | 80.49 |
| | T-Patcher+ODEdit | **81.85** | **80.47** | 86.25 | **85.37** | **73.44** | **74.28** | 71.18 | **81.67** |
| | UniKE | 94.32 | 87.18 | 95.98 | 93.15 | 74.01 | 73.84 | 76.09 | 82.36 |
| | UniKE+ODEdit | **96.58** | **89.34** | **96.17** | **93.27** | **74.52** | **75.49** | **76.65** | **83.28** |
| | Pre-edited | 19.21 | 24.08 | 99.44 | 91.56 | 0 | 0 | 99.79 | 94.93 |
| MiniGPT-4 7B | FT | 100 | 100 | 97.50 | 40.85 | 100 | 0 | 95.00 | 39.83 |
| | IKE | 99.95 | 99.90 | 50.02 | 3.31 | 90.30 | 90.00 | 51.49 | 4.27 |
| | SERAC | 91.70 | 98.60 | 99.99 | 3.72 | 83.60 | 93.10 | 99.99 | 4.65 |
| | WISE | 100 | 100 | 90.10 | 52.15 | 100 | 91.58 | 92.81 | 70.68 |
| | WISE+ODEdit | **100** | 97.50 | **92.39** | **62.14** | **100** | 90.04 | **94.54** | **73.17** |
| | MEND | 96.20 | 96.00 | 99.42 | 88.25 | 77.80 | 74.60 | 99.28 | 87.85 |
| | MEND+ODEdit | **97.00** | **97.00** | **99.52** | **88.61** | **78.60** | 74.20 | **99.36** | 86.77 |
| | T-Patcher | 70.56 | 68.79 | 64.45 | 81.77 | 69.54 | 68.95 | 63.59 | 81.34 |
| | T-Patcher+ODEdit | **72.38** | **72.11** | **65.29** | **82.93** | **71.42** | **70.98** | **65.03** | **82.75** |
| | UniKE | 84.32 | 81.29 | 78.45 | 85.81 | 72.18 | 70.41 | 68.53 | 84.59 |
| | UniKE+ODEdit | **85.14** | **83.23** | **79.35** | **86.56** | **73.06** | **71.58** | **69.46** | **85.12** |

Table 2: **Results of long-term editing on BLIP2-OPT.**

| Dataset | Model | T=5 | | | | T=10 | | | |
|---|---|---|---|---|---|---|---|---|---|
| | | Rel.↑ | Gen.↑ | T-Loc.↑ | M-Loc.↑ | Rel.↑ | Gen.↑ | T-Loc.↑ | M-Loc.↑ |
| E-VQA | WISE | 44.50 | 34.75 | 0.40 | 0.15 | 28.50 | 24.55 | 0.63 | 0.15 |
| | WISE+ODEdit | 49.42 | 43.52 | 0.80 | 0.15 | 43.33 | 24.22 | 0.81 | 0.15 |
| | UniKE | 90.28 | 80.26 | 91.41 | 89.37 | 86.52 | 76.58 | 87.64 | 86.31 |
| | UniKE+ODEdit | 92.63 | 83.59 | 92.38 | 89.95 | 89.79 | 81.25 | 89.35 | 87.54 |
| E-IC | WISE | 84.31 | 65.49 | 0.76 | 0.14 | 75.96 | 55.56 | 0.71 | 0.14 |
| | WISE+ODEdit | 86.53 | 66.94 | 0.94 | 0.14 | 84.64 | 61.70 | 0.77 | 0.14 |
| | UniKE | 70.16 | 71.45 | 72.09 | 79.52 | 63.54 | 64.71 | 66.29 | 73.25 |
| | UniKE+ODEdit | 71.05 | 73.22 | 72.68 | 80.77 | 65.87 | 68.82 | 67.11 | 76.59 |

**ODEdit consistently mitigates this decay and enhances stability. By learning invariant trajectories, ODEdit preserves higher reliability, generality, and locality, and the improvement becomes more pronounced as $T$ increases. These results prove the ability of ODEdit to discern and stabilize core causal features against the variations introduced by successive edits.**

### 4.4 IN-DEPTH ANALYSIS

**Ablations of OOD Risks.** We conduct ablation studies by removing each risk separately. The results in Table 3 show: 1) **Reliability risk is essential for knowledge assimilation.** The removal of $\mathcal{R}_{rel}$ causes Rel. and Gen. to collapse, showing MLLM fails to learn the intended knowledge. 2) **Locality risk constrains editing within scope.** Ablating $\mathcal{R}_{loc}$ leads to a decrease in T-Loc. and M-Loc., indicating edit effects spill over into irrelevant knowledge areas, causing causal-overfit and violating locality. **3) Generality risk facilitates semantic generalization. The addition of $\mathcal{R}gen$ yields a substantial gain in the Gen. metric (e.g., from 86.59 to 89.34 on E-VQA and from 71.24 to 75.49 on E-IC). While it induces minimal fluctuations in Rel. and minor variations in Loc., this aligns with our OOD formulation where the three risks exhibit inherent cross-effects. $\mathcal{R}gen$**

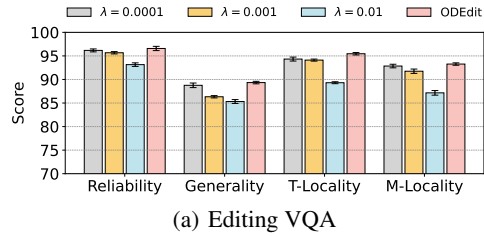 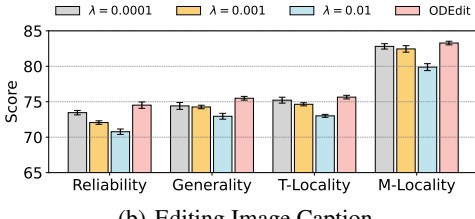

| (a) Editing VQA | (b) Editing Image Caption |
|:---:|:---:|

Figure 2: Results of ablation study to illustrate the effect of IRM-TV optimization.

Table 3: Results of ablation study to illustrate effects of each OOD risk.

| Invariants | Editing VQA (E-VQA) | | | | Editing Image Caption (E-IC) | | | |
|---|---|---|---|---|---|---|---|---|
| | Rel.↑ | Gen.↑ | T-Loc.↑ | M-Loc.↑ | Rel.↑ | Gen.↑ | T-Loc.↑ | M-Loc.↑ |
| w/o $\mathcal{R}_{rel}$ | 0 | 0 | 99.85 | 98.63 | 0 | 0 | 95.81 | 97.22 |
| w/o $\mathcal{R}_{loc}$ | 96.65 | 89.51 | 74.36 | 71.25 | 74.62 | 75.45 | 64.49 | 73.19 |
| w/o $\mathcal{R}_{gen}$ | 96.49 | 86.59 | 95.97 | 93.54 | 74.60 | 71.24 | 75.83 | 83.60 |
| ODEdit | 96.58 | 89.34 | 95.46 | 93.27 | 74.52 | 75.49 | 75.65 | 83.28 |

primarily works by aligning semantic-neighboring samples with the edited instance in the latent space, which successfully promotes invariant feature learning to prevent causal underfit. The slight impact on locality can be attributed to the potential reinforcement of local concept-output associations as a byproduct of this semantic alignment process.

**Effects of Maximum Mean Discrepancy Alignment.** We perform ablations with invariants: (a) *MMD-s RBF* denotes MMD with a Radial Basis Function (RBF) kernel and a single rephrase prompt. (b) *MMD-s Linear* with linear kernel. (c) *MMD-m RBF* utilizes multiple rephrase prompts. (d) *Contrast* replace MMD with contrastive learning. Results in Table 4 show that *MMD-s RBF* achieves the most balanced and effective performance. *MMD-s Linear* is less effective at capturing cross-modal semantic distributions. The gap between *Contrast* and *MMD-s RBF* underscores advantages of a distribution-level alignment objective over instance-level. An insightful finding is that using multiple rephrase prompts yields no additional benefit. The potential reason is that a single rephrase prompt provides a focused semantic transformation path, while multiple prompts introduce noisy variations which might lead to spurious correlations.

Table 4: Ablation studies on the MMD alignment.

| Invariant | Rel. | Gen. | T-Loc. | M-Loc. |
|---|---|---|---|---|
| MMD-s RBF | 79.40 | 64.40 | 99.01 | 86.14 |
| MMD-s Linear | 78.80 | 63.20 | 98.49 | 80.75 |
| MMD-m RBF | 76.81 | 63.59 | 99.00 | 85.73 |
| Contrast | 76.40 | 63.80 | 99.00 | 89.91 |

**Performance on other MLLMs.** We further conduct editing on other MLLMs, *i.e.,* LLaVA (Liu et al., 2023). From Table 5, ODEdit enhances WISE across all metrics on LLaVA, with particularly notable gains in generality and locality. These robust improvements on a distinct MLLM architecture underscore the strong generalizability of ODEdit, which stems from its core design of learning invariant editing trajectories that effectively suppress spurious correlations across diverse model backbones.

Table 5: Results on other MLLMs.

| E-VQA | Rel. | Gen. | T-Loc. | M-Loc. |
|---|---|---|---|---|
| WISE | 100 | 71.42 | 91.51 | 93.75 |
| WISE+ODEdit | 100 | 72.01 | 94.40 | 95.47 |
| **E-IC** | **Rel.** | **Gen.** | **T-Loc.** | **M-Loc.** |
| WISE | 99.89 | 81.28 | 92.69 | 94.63 |
| WISE+ODEdit | 99.78 | 82.22 | 92.54 | 95.92 |

**Effects of Edit Trajectory Invariant Learning.** We ablate the effect of the TV-$\ell_1$ penalty strength ($\lambda$) in IRM-TV optimization, and present results in Figure 2, from which we find: 1) An insufficient penalty, *i.e.,* $\lambda = 0.0001$, fails to extract feature invariance, thus hurting generality extremely. 2) An excessive penalty, *i.e.,* $\lambda = 0.01$, over-constrains the model, simultaneously degrading three metrics. An overly strong invariance constraint makes the model's internal representations rigid, so that failing to make updates to target knowledge while incorrectly altering peripheral features it should preserve. 3) The dynamic and adaptive formulation of $\lambda(\pi, \phi_e)$ shows its superiority, validating it robustly balances knowledge assimilation and discrimination across diverse environments.

**Visualization on OOD Generalization.** We visualize the latent representations of original and rephrased prompts in MLLM with t-SNE (Van der Maaten & Hinton, 2008) across dif-

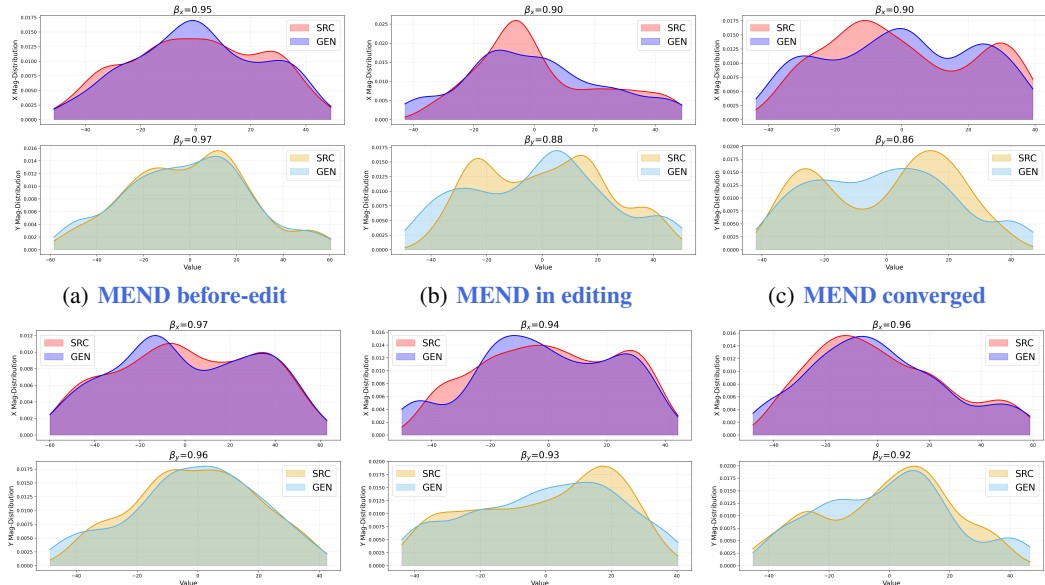

(a) **MEND before-edit**  (b) **MEND in editing**  (c) **MEND converged**

(d) **MEND+ODEdit before-edit**  (e) **MEND + ODEdit in editing**  (f) **MEND + ODEdit converged**

Figure 3: **The t-SNE distributions of the latent representations on original prompts (SRC) and rephrase prompts (GEN) in MLLM. The curves depict the marginal distributions along the two dimensions, with $\beta_x$ and $\beta_y$ representing the proportion of the overlap.**

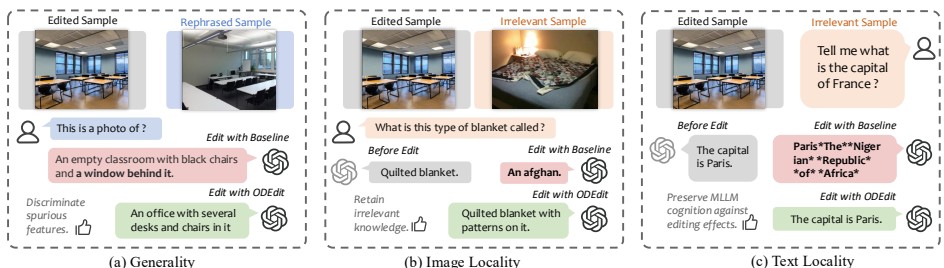

(a) Generality  (b) Image Locality  (c) Text Locality

Figure 4: Case studies on the evaluation for generality, image locality, and text locality.

**ferent editing stages. From Figure 3: 1) Before editing, rephrased prompts align with original prompt distributions in the pre-trained MLLM. 2) During editing, MEND induces a marked distribution shift as $\beta_x$ and $\beta_y$ values drop, fails to extract semantic invariance. But MEND+ODEdit maintains strong alignment with high $\beta$ values, showing stable trajectory learning. 3) At Convergence, the distribution shift in MEND persists while ODEdit sustains robust alignment, proving superior generalization to semantic-neighboring regions.**

**Hyperparameter Sensitivity.** We study effects of the learning rate and layer depth in the IRM-TV network. From Figure 5: 1) A small learning rate hinders extraction of invariant features, while a moderate increase enhances generality, accompanied by a slight sacrifice in locality. However, an excessively large rate suppresses overall performance. 2) Deeper networks facil-

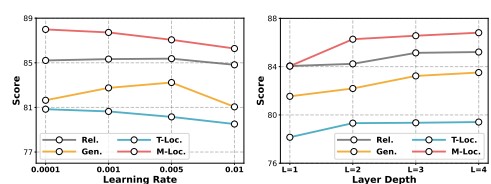

Figure 5: Effects of learning rate and layer depth.

itate diverse cross-modal association learning, but the marginal benefit diminishes once layer depth reaches a certain level. **Principled guidelines for setting parameters in Appendix D.5.**

**Computational Cost. We pick one typical parameter-adjusting (MEND) and model-extending (WISE) baselines for comparison. From Table 6: 1) ODEdit introduces a supplementary network that processes parameters from the knowledge-editing layer, leading to increased memory usage. But this is the reasonable trade-off for the gains in performance and is acceptable given the current state of computational resources. 2) ODEdit does not incur a significant increase in time cost, indicating its efficiency. 3) While integrating ODEdit increases steps,**

Table 6: **Computational cost comparison on E-IC. Memo. = Memory usage, Edit-T/sp = Editing time per step, Train-T/sp = Training time per step, All-T/sp = Total time per step.**

| Models | BLIP2-OPT | | | | | MiniGPT-4 | | | | |
|---|---|---|---|---|---|---|---|---|---|---|
| | Memo. | Edit-T/sp | Train-T/sp | All-T/sp | Steps. | Memo. | Edit-T/sp | Train-T/sp | All-T/sp | Steps. |
| WISE | 28.47GB | 0.401 | 0.023 | 0.424 | 7 | 36.59GB | 0.473 | 0.035 | 0.508 | 8 |
| WISE+ODEdit | 47.53GB | 0.442 | 0.043 | 0.485 | 7 | 69.36GB | 0.533 | 0.072 | 0.605 | 8 |
| MEND | 14.75GB | 1.369 | 0.018 | 1.387 | 25000 | 25.30GB | 1.676 | 0.188 | 1.865 | 10000 |
| MEND+ODEdit | 36.05GB | 1.480 | 0.116 | 1.596 | 45000 | 62.80GB | 1.861 | 0.204 | 2.066 | 15000 |

**the resultant increase in total time cost does not constitute an order-of-magnitude change and remains within a practical range for real-world deployment. Employing higher-performance computing resources would substantially reduce this training time gap.**

**Interpretability Studies.** Qualitative cases in Figure 4 and more analysis in Appendix D.6.

## 5 RELATED WORK

**Unimodal LLM Editing.** Model editing aims to modify the target knowledge in LLM while preserving irrelevant concepts. Previous approaches can be divided into two types. *Parameter-adjusting methods* modify intrinsic parameters of LLMs to update new knowledge. In this line, locate-then-edit models such as ROME (Meng et al., 2022a), MEMIT (Meng et al., 2022b), GLAME (Zhang et al., 2024b), AnyEdit (Jiang et al., 2025), AlphaEdit (Fang et al., 2024), first identify crucial knowledge-related parameters and then perform targeted edits. Besides, meta-learning based approaches like KE (De Cao et al., 2021), MEND (Mitchell et al., 2021), InstructEdit (Zhang et al., 2024c), determine parameter modifications by training hypernetworks. Contrastingly, *model-extending methods* incorporate additional components to store new knowledge while keeping original model parameters. The added components take diverse forms, including memory in SERAC (Mitchell et al., 2022) and WISE (Wang et al., 2024a), auxiliary neurons in T-Patcher (Huang et al., 2023), codebooks in GRACE (Hartvigsen et al., 2023), and LoRA modules in MELO (Yu et al., 2024). Other works like MemPrompt (Madaan et al., 2022), IKE (Zheng et al., 2023), and DeCK (Bi et al., 2024) utilize in-context learning to update factual knowledge. Despite their efficacy in unimodal LLM editing, they suffer from causal-underfit and causal-overfit issues in MLLM.

**Multimodal LLM Editing.** Recent advances in MLLMs (Li et al., 2023; 2024a; Ma et al., 2025) have motivated research on multimodal knowledge editing (Pan et al., 2023; Zhou et al.). A series of benchmarks, *e.g.,* MMEdit (Cheng et al., 2023), MIKE (Li et al., 2024b), VLKEB (Huang et al., 2024), MC-MKE (Zhang et al., 2024a), MMKE (Du et al., 2025), provide unified datasets and evaluation to assess multimodal editing efficacy. However, research on strengthening the robustness of MLLM editing methods holistically across reliability, locality, and generality remains under-explored. MSCKE (Zeng et al., 2024) establishes a multimodal scope classifier-based knowledge editor to identify and update specific visual entities. UniKE (Pan et al., 2024) integrates intrinsic knowledge editing and external knowledge resorting to promote locality and generality. **BalancEdit (Guo et al., 2025) performs codebook-based edits that balance generality and locality by using contrastive samples to localize each fact's influence.** Nevertheless, existing work remains constrained to rigid parameter-to-output mappings, which prevent MLLMs from intelligently distinguishing between semantic-shift and factual-shift, thereby hindering adaptive and robust editing.

**Out-of-Distribution Generalization.** We present related work in this field in Appendix E.

## 6 CONCLUSION AND FUTURE WORK

In this work, we rethink knowledge editing in MLLM as an OOD generalization problem. To identify semantic-shift and factual-shift among various cross-modal prompting environments, we propose a plug-and-play invariant learning based optimization paradigm with tripartite OOD risks to jointly enhance editing reliability, locality, and generality. This work marks an initial step in solving multimodal editing from an OOD perspective, for which we introduce simple yet general editing invariant risk metrics with an pathway to guide robust model adaptation. In the future, researchers could investigate advanced strategies to strengthen MLLM's grasp of invariant trajectories and discern spurious factors, with refined regularization functions for more robust cross-modal editing.

## ETHICS STATEMENT

This work adheres to the ICLR Code of Ethics. All experiments are conducted on publicly available datasets without involving any personally identifiable or sensitive user information. No human subjects were recruited, and no private data was collected or released. We are not aware of any ethical concerns or potential risks associated with the deployment of our approach.

## REPRODUCIBILITY STATEMENT

To ensure the reproducibility of our work, we have taken the following steps. The source code for ODEdit, including implementations of the tripartite OOD risk and the Edit Trajectory Invariant Learning algorithm, has been made publicly available at `https://anonymous.4open.science/r/ODEdit-2756`. Complete theoretical proofs for our key propositions, including the equivalence between the OOD and IRM-TV objectives, are provided in Appendix A. Detailed descriptions of the experimental setup, including the MLLM backbones (Appendix B.1), baseline methods (Appendix B.2), hyperparameter configurations, and training procedures, are thoroughly documented in Appendix B.3. The MMEdit benchmark used for evaluation is publicly available, and our data processing steps are clearly outlined in Section 4.1 and Appendix B. We hope these resources will facilitate the replication and extension of our work.

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

# A    PROOFS

## A.1    EQUIVALENCE BETWEEN OOD-w AND IRM FORMULATION

In this section, we provide a detailed proof of the equivalence between the original out-of-distribution (OOD) editing objective and its reformulation using an environment-aware classifier $\omega$. Specifically, we aim to show that for any edited model parameters $\phi_e$, the worst-case risk over all environments $e \in \mathcal{E}$ can be equivalently expressed as the worst-case risk over all possible classifiers $\omega \in \Sigma$, under the condition that $\omega$ captures the environmental variability through a surjective mapping.

**Proof.** To establish the equality, we demonstrate two inequalities. First, we prove that

$$\max_{e \in \mathcal{E}} \mathcal{R}_{edit}(\omega(e) \circ \phi_e) \geq \max_{\omega \in \Sigma} \mathcal{R}_{edit}(\omega \circ \phi_e).$$

Let $\omega^*$ be a classifier that attains the maximum on the right-hand side, so that

$$\omega^* = \arg \max_{\omega \in \Sigma} \mathcal{R}_{edit}(\omega \circ \phi_e).$$

Given the surjectivity of the mapping $e \mapsto \omega(e)$, there exists an environment $e_0 \in \mathcal{E}$ such that $\omega(e_0) = \omega^*$. Consequently,

$$\mathcal{R}_{edit}(\omega(e_0) \circ \phi_e) = \mathcal{R}_{edit}(\omega^* \circ \phi_e) = \max_{\omega \in \Sigma} \mathcal{R}_{edit}(\omega \circ \phi_e).$$

Since $e_0$ is an element of $\mathcal{E}$, the maximum over $\mathcal{E}$ must be at least as large as the value at $e_0$, yielding the desired inequality. Second, we prove the opposite inequality:

$$\max_{e \in \mathcal{E}} \mathcal{R}_{edit}(\omega(e) \circ \phi_e) \leq \max_{\omega \in \Sigma} \mathcal{R}_{edit}(\omega \circ \phi_e).$$

Let $e^*$ be an environment that achieves the maximum on the left-hand side, i.e.,

$$e^* = \arg \max_{e \in \mathcal{E}} \mathcal{R}_{edit}(\omega(e) \circ \phi_e).$$

Then, $\omega(e^*)$ is by construction a member of $\Sigma$. Therefore,

$$\mathcal{R}_{edit}(\omega(e^*) \circ \phi_e) \leq \max_{\omega \in \Sigma} \mathcal{R}_{edit}(\omega \circ \phi_e),$$

which directly implies the inequality. By combining both inequalities, we conclude that

$$\max_{e \in \mathcal{E}} \mathcal{R}_{edit}(\omega(e) \circ \phi_e) = \max_{\omega \in \Sigma} \mathcal{R}_{edit}(\omega \circ \phi_e),$$

which holds for any $\phi_e$. Thus, minimizing either expression with respect to $\phi_e$ leads to the same optimal solution, confirming the equivalence between the OOD-$\omega$ and IRM formulations. This result allows us to leverage the IRM framework for invariant learning in multimodal knowledge editing.

## A.2    HOW IRM WITH TV-$l_1$ PENALTY ACHIEVES EDITING OOD

In this section, we provide a theoretical analysis demonstrating how the proposed IRM formulation with TV-$\ell_1$ penalty achieves the out-of-distribution (OOD) editing objective defined in Eq.11 of the main text. Specifically, we prove that when the penalty parameter $\lambda_{\phi_e}$ is allowed to vary with the model parameters $\phi_e$, the IRM-TV objective can achieve the same optimum as the original OOD editing objective. Recall the IRM-TV formulation from Eq.11:

$$\min_{\phi_e} \left\{ \mathbb{E}_\omega[\mathcal{R}_{rel}(\omega \circ \phi_e) + \mathcal{R}_{loc}(\omega \circ \phi_e)] + \lambda_{\phi_e} \left( \mathbb{E}_\omega[|\nabla_\omega \mathcal{R}_{gen}(\omega \circ \phi_e)|] \right)^2 \right\},$$

where the first term represents the base editing risk (reliability + locality) and the second term is the TV-$\ell_1$ penalty on the generalization risk. The original OOD editing objective is:

$$\min_{\phi_e} \max_{\omega \in \Sigma} \mathcal{R}_{edit}(\omega \circ \phi_e).$$

We first demonstrate through a counterexample that a fixed $\lambda$ cannot achieve the OOD objective, then prove the existence of a $\lambda_{\phi_e}$ that varies with $\phi_e$ to achieve equivalence.

### A.2.1 The Necessity of $\lambda$ Varying with $\phi_e$

Following (Lai & Wang, 2024), we provide a counterexample with a fixed $\lambda$ to prove the necessity of $\lambda$ Varying with $\phi_e$. To show that a fixed $\lambda$ is insufficient, consider a simplified editing scenario where we aim to optimize the feature parameter $\phi_e \in [-1, 1]$. The classifier $\omega$ follows a uniform distribution on $[-0.9, 0.1]$, reflecting environmental variations. The editing risk is defined as:

$$\mathcal{R}_{edit}(\omega \circ \phi_e) := |\omega \cdot \phi_e + 1|.$$

For this setup, the OOD objective achieves its minimum at $\phi_e = 0$ with value 1:

$$\min_{\phi_e \in [-1,1]} \max_{\omega \in [-0.9, 0.1]} \mathcal{R}_{edit}(\omega \circ \phi_e) = 1,$$

$$\arg \min_{\phi_e \in [-1,1]} \max_{\omega \in [-0.9, 0.1]} \mathcal{R}_{edit}(\omega \circ \phi_e) = 0.$$

However, for any fixed $\lambda \geq 0$, the IRM-TV objective:

$$\mathbb{E}_\omega[\mathcal{R}_{edit}(\omega \circ \phi_e)] + \lambda \left( \mathbb{E}_\omega[|\nabla_\omega \mathcal{R}_{edit}(\omega \circ \phi_e)|] \right)^2,$$

fails to achieve the same optimum as the OOD objective. To demonstrate this, we analyze the behavior of both objectives for the simplified editing scenario. For $\phi_e \geq 0$, the OOD objective becomes $\max_{\omega \in [-0.9, 0.1]} \mathcal{R}_{edit}(\omega \circ \phi_e) = 1 + 0.1\phi_e$, which is minimized at $\phi_e = 0$ with value 1. For $\phi_e < 0$, the OOD objective becomes $\max_{\omega \in [-0.9, 0.1]} \mathcal{R}_{edit}(\omega \circ \phi_e) = 1 - 0.9\phi_e$, which is also minimized at $\phi_e = 0$ with value 1. Now, evaluating the IRM-TV objective with fixed $\lambda$:

$$\mathbb{E}_\omega[\mathcal{R}_{edit}(\omega \circ \phi_e)] = \int_{-0.9}^{0.1} (1 + \omega\phi_e) d\nu = 1 + \phi_e \cdot \mathbb{E}_\omega[\omega] = 1 - 0.4\phi_e,$$

$$\mathbb{E}_\omega[|\nabla_\omega \mathcal{R}_{edit}(\omega \circ \phi_e)|] = |\phi_e| \cdot \int_{-0.9}^{0.1} d\nu = |\phi_e|.$$

Thus, the IRM-TV objective becomes $1 - 0.4\phi_e + \lambda\phi_e^2$. Minimizing this quadratic function over $\phi_e \in [-1, 1]$ yields: 1) If $\lambda > 0.2$, the minimum occurs at $\phi_e = 0.2/\lambda$ with value $1 - 0.04/\lambda$. 2) If $0 < \lambda \leq 0.2$, the minimum occurs at $\phi_e = 1$ with value $0.6 + \lambda$. 3) If $\lambda = 0$, the minimum occurs at $\phi_e = 1$ with value 0.6. Comparing with the OOD optimum ($\phi_e = 0$, value 1), for any fixed $\lambda \geq 0$ we observe that:

$$\min_{\phi_e \in [-1,1]} \left\{ 1 - 0.4\phi_e + \lambda\phi_e^2 \right\} \neq 1,$$

$$\arg \min_{\phi_e \in [-1,1]} \left\{ 1 - 0.4\phi_e + \lambda\phi_e^2 \right\} \neq 0.$$

This deviation occurs because the expectation term $\mathbb{E}_\omega[\mathcal{R}_{edit}(\omega \circ \phi_e)]$ pulls the optimum away from $\phi_e = 0$ to reduce the average risk, while the fixed $\lambda$ cannot adequately compensate for this bias. Only when $\lambda$ is allowed to vary with $\phi_e$ can we achieve equivalence with the OOD objective.

### A.2.2 Proofs on Existence of $\lambda_{\phi_e}$

We now prove that there exists a $\lambda_{\phi_e}$ that varies with $\phi_e$ such that the IRM-TV objective equals the OOD objective for each $\phi_e$. For the case where $\mathbb{E}_\omega[|\nabla_\omega \mathcal{R}_{edit}(\omega \circ \phi_e)|] = 0$, indicating constant generalization risk, $\lambda_{\phi_e}$ can be chosen arbitrarily since the TV term vanishes. For the nontrivial case where $\mathbb{E}_\omega[|\nabla_\omega \mathcal{R}_{gen}(\omega \circ \phi_e)|] > 0$, we construct $\lambda_{\phi_e}$ as:

$$\lambda_{\phi_e} := \frac{\max_{\omega \in \Sigma} \mathcal{R}_{edit}(\omega \circ \phi_e) - \mathbb{E}_\omega[\mathcal{R}_{rel}(\omega \circ \phi_e) + \mathcal{R}_{loc}(\omega \circ \phi_e) + \mathcal{R}_{gen}(\omega \circ \phi_e)]}{\left( \mathbb{E}_\omega[|\nabla_\omega \mathcal{R}_{edit}(\omega \circ \phi_e)|] \right)^2}.$$

This construction ensures that for each $\phi_e$, we have

$$\mathbb{E}_\omega[\mathcal{R}_{edit}(\omega \circ \phi_e)] + \lambda_{\phi_e} \left( \mathbb{E}_\omega[|\nabla_\omega \mathcal{R}_{edit}(\omega \circ \phi_e)|] \right)^2 = \max_{\omega \in \Sigma} \mathcal{R}_{edit}(\omega \circ \phi_e),$$

since the numerator represents the gap between the worst-case risk and the expected base risk.

### A.2.3 ACHIEVING OOD-$\omega$ OPTIMALITY

Let $\phi_e^*$ be an optimal solution of the IRM-TV objective with $\lambda_{\phi_e}$ defined above. Then for any $\phi_e$:

$$\mathbb{E}_\omega[\mathcal{R}_{rel}(\omega \circ \phi_e^*) + \mathcal{R}_{loc}(\omega \circ \phi_e^*) + \mathcal{R}_{gen}(\omega \circ \phi_e^*)] + \lambda_{\phi_e^*} \left(\mathbb{E}_\omega[|\nabla_\omega \mathcal{R}_{edit}(\omega \circ \phi_e^*)|]\right)^2$$

$$\leq \mathbb{E}_\omega[\mathcal{R}_{rel}(\omega \circ \phi_e) + \mathcal{R}_{loc}(\omega \circ \phi_e) + \mathcal{R}_{gen}(\omega \circ \phi_e)] + \lambda_{\phi_e} \left(\mathbb{E}_\omega[|\nabla_\omega \mathcal{R}_{edit}(\omega \circ \phi_e)|]\right)^2.$$

Substituting the definition of $\lambda_{\phi_e}$, for all $\phi_e$ we have:

$$\max_{\omega \in \Sigma} \mathcal{R}_{edit}(\omega \circ \phi_e^*) \leq \max_{\omega \in \Sigma} \mathcal{R}_{edit}(\omega \circ \phi_e).$$

This is the proof that $\phi_e^*$ is also optimal for the OOD objective. Conversely, if $\phi_e^*$ is optimal for the OOD objective, then for all $\phi_e$:

$$\mathbb{E}_\omega[\mathcal{R}_{edit}(\omega \circ \phi_e^*)] + \lambda_{\phi_e^*} \left(\mathbb{E}_\omega[|\nabla_\omega \mathcal{R}_{edit}(\omega \circ \phi_e^*)|]\right)^2 = \max_{\omega \in \Sigma} \mathcal{R}_{edit}(\omega \circ \phi_e^*) \leq \max_{\omega \in \Sigma} \mathcal{R}_{edit}(\omega \circ \phi_e),$$

which shows that $\phi_e^*$ is also optimal for the IRM-TV objective. This completes the proof that the IRM formulation with TV-$\ell_1$ penalty can achieve the OOD editing objective when $\lambda_{\phi_e}$ is properly chosen as a function of $\phi_e$.

### A.3 COMPUTATION OF GRADIENTS IN PRIMAL-DUAL OPTIMIZATION

In this section, we provide the detailed computation process of the gradient $\nabla_\delta \mathcal{G}$ and the subgradient $\partial_{\phi_e} \mathcal{G}$ for the primal-dual optimization problem defined in Eq. 12 and 13 of the main text. Recall the Lagrangian function as:

$$\mathcal{G}(\delta, \phi_e) = \mathbb{E}_\omega[\mathcal{R}_{\text{edit}}(\omega \circ \phi_e)] + \lambda(\delta, \phi_e) \left(\mathbb{E}_\omega[|\nabla_\omega \mathcal{R}_{\text{edit}}(\omega \circ \phi_e)|]\right)^2,$$

where $\mathcal{R}_{\text{edit}}(\omega \circ \phi_e) = \mathcal{R}_{\text{rel}}(\omega \circ \phi_e) + \mathcal{R}_{\text{loc}}(\omega \circ \phi_e) + \mathcal{R}_{\text{gen}}(\omega \circ \phi_e)$ represents the complete editing risk. To compute the gradients, we assume that the risk functions are Lipschitz continuous and admit subgradients at non-differentiable points.

**Subgradient of $\mathcal{G}$ with Respect to $\phi_e$.** The subgradient $\partial_{\phi_e} \mathcal{G}(\delta, \phi_e)$ is computed as:

$$\partial_{\phi_e} \mathcal{G}(\delta, \phi_e) = \mathbb{E}_\omega[\nabla_{\phi_e} \mathcal{R}_{\text{edit}}(\omega \circ \phi_e)] + 2\lambda(\delta, \phi_e) \cdot \mathbb{E}_\omega[|\nabla_\omega \mathcal{R}_{\text{edit}}(\omega \circ \phi_e)|]$$

$$\cdot \mathbb{E}_\omega[\partial_{\phi_e}|\nabla_\omega \mathcal{R}_{\text{edit}}(\omega \circ \phi_e)|] + \nabla_{\phi_e} \lambda(\delta, \phi_e) \cdot \left(\mathbb{E}_\omega[|\nabla_\omega \mathcal{R}_{\text{edit}}(\omega \circ \phi_e)|]\right)^2.$$

Here, the term $\partial_{\phi_e}|\nabla_\omega \mathcal{R}_{\text{edit}}(\omega \circ \phi_e)|$ requires special handling due to the absolute value function. Based on derivations in (Wang et al., 2025), we obtain its subgradient as:

$$\partial_{\phi_e}|\nabla_\omega \mathcal{R}_{\text{edit}}(\omega \circ \phi_e)| = \begin{cases} \text{sign}(\nabla_\omega \mathcal{R}_{\text{edit}}(\omega \circ \phi_e)) J_{\phi_e}^{-1} \left[\nabla_\omega \mathcal{R}_{\text{edit}}(\omega \circ \phi_e)\right] & \text{if } \nabla_\omega \mathcal{R}_{\text{edit}}(\omega \circ \phi_e) \neq 0, \\ 0 & \text{if } \nabla_\omega \mathcal{R}_{\text{edit}}(\omega \circ \phi_e) = 0, \end{cases}$$

where $J_{\phi_e}[\cdot]$ denotes the Jacobian matrix with respect to $\phi_e$. This formulation ensures that the subgradient is well-defined even at points where the gradient is zero.

**Gradient of $\mathcal{G}$ with Respect to $\delta$.** The gradient $\nabla_\delta \mathcal{G}(\delta, \phi_e)$ is computed as:

$$\nabla_\delta \mathcal{G}(\delta, \phi_e) = \nabla_\delta \lambda(\delta, \phi_e) \cdot \left(\mathbb{E}_\omega[|\nabla_\omega \mathcal{R}_{\text{edit}}(\omega \circ \phi_e)|]\right)^2.$$

The first term in $\mathcal{G}$, $\mathbb{E}_\omega[\mathcal{R}_{\text{edit}}(\omega \circ \phi_e)]$, does not depend on $\delta$, so its gradient with respect to $\delta$ is zero.

**Implementation Notes.** In practice, the expectations over $\omega$ are approximated using Monte Carlo sampling from the environment distribution. The gradients $\nabla_{\phi_e} \mathcal{R}_{\text{edit}}$, $\nabla_\omega \mathcal{R}_{\text{edit}}$, and $\nabla_\delta \lambda$ are computed using standard backpropagation. The subgradient for the absolute value term is implemented using a conditional statement, which is supported by autograd systems. This approach ensures efficient and stable optimization during the primal-dual updates.

These gradient computations enable the iterative updates in Eq. 14 of the main text:

$$\phi_e^{(k+1)} = \phi_e^{(k)} - \gamma_1^{(k)} \cdot \partial_{\phi_e} \mathcal{G}(\delta^{(k)}, \phi_e^{(k)}), \quad \delta^{(k+1)} = \delta^{(k)} + \gamma_2^{(k)} \cdot \nabla_\delta \mathcal{G}(\delta^{(k)}, \phi_e^{(k+1)}),$$

leading to convergence to a solution that minimizes the OOD editing risk while maintaining the invariance properties enforced by the TV-$\ell_1$ penalty.

## B  CAUSAL GROUNDING ANALYSIS OF CASCADED REASONING IN MLLM

### B.1  ARCHITECTURAL CAUSAL STRUCTURE EMBEDDED IN MLLM

Multimodal language models implement an unidirectional computational graph, that is: unimodal encoders $\rightarrow$ cross-modal fusion $\rightarrow$ unified semantic reasoning. This forward computation defines a structural causal ordering, for which in the Structural Causal Model (SCM) view (Li et al., 2024c; Zhou et al., 2024), modules are equal to variables and the forward pass is equal to structural equations. Thus the cascade reasoning is not a hypothesized causal model, but the deterministic functional decomposition of existing architectures.

### B.2  HOW PERTURBATIONS PROPAGATE DURING MLLM EDITING

Under this structural ordering, any local perturbation to a module $\Delta M$ or parameter $\Delta W$ necessarily propagates forward through the downstream modules and changes their internal states. Thus, there is no one-to-one rigid mapping, *i.e.*, rigid mapping, between a specific parameter edit and the final output change, because the effect of the edit is mediated by all subsequent causal mechanisms in the network. Formally, for a structural chain

$$h^{(unimodal)} \rightarrow h^{(align)} \rightarrow h^{(shared)} \rightarrow y$$

A perturbation enters the output through

$$y' = f_{shared}\left(f_{align}\left(f_{unimodal}(x; W + \delta W)\right)\right)$$

Thus the output shift $\Delta y$ depends not only on $\Delta W$, but on how $\Delta W$ perturbs $h_{unimodal}$, how this shifted representation perturbs $h_{align}$, and subsequently how the changed alignment influences the semantic reasoning module $h_{shared}$. This cascading mediation proves why treating *parameter edit $\rightarrow$ output change* as a rigid mapping is fundamentally inaccurate in MLLMs, *i.e.,* cause *casual underfit* and *casual overfit* in Section Introduction.

## C  DEFINITIONS OF SEMANTIC SHIFT AND FACTUAL SHIFT

The definitions of *Semantic Shift* and *Factual Shift* rely on three shared mappings:

**Semantic neighborhood.**  Let $f(x)$ be the MLLM's semantic embedding. We define meaning-preserving variation via the semantic neighborhood:

$$\mathcal{N}_\varepsilon(x) = \{x' : \|f(x') - f(x)\|_2 \leq \varepsilon\}.$$

**Atomic factual content.**  Let $k(x)$ denote the atomic factual content (e.g., entity–attribute or entity–relation tuples). Two inputs share factual content iff $k(x) = k(x')$.

**Output-relevant concept mapping.**  Let $c(x)$ denote the minimal set of conceptual factors that feed into the MLLM's forward causal chain (perception $\rightarrow$ alignment $\rightarrow$ semantic reasoning) and determine the final output:

$$y = MLLM(c(x)).$$

**Definition 1** (Semantic Shift). *A sample $x'$ exhibits semantic shift w.r.t. $x$ if and only if*

$$x' \in \mathcal{N}_\varepsilon(x), \quad k(x') = k(x), \quad c(x) \cap c(x') \neq \varnothing, \quad \mathrm{MLLM}\left(c(x')\right) = \mathrm{MLLM}(c(x))$$

That is, semantic shift refers to variations within the semantic neighborhood while preserving factual content and preserving the output-relevant conceptual factors. Typical examples include paraphrases, lexical substitutions, stylistic rewordings, and mild visual variations.

**Definition 2** (Factual Shift). *To be rigorous, there should be two kinds of factual shift, i.e., easy factual shift and hard factual shift:*

*Easy Factual Shift:*  $x' \notin \mathcal{N}_\varepsilon(x), k(x') \neq k(x), c(x) \cap c(x') = \varnothing, \mathrm{MLLM}\left(c(x')\right) \neq \mathrm{MLLM}(c(x))$

*Hard Factual Shift:*  $x' \notin \mathcal{N}_\varepsilon(x), k(x') \neq k(x), c(x) \cap c(x') \neq \varnothing, \mathrm{MLLM}\left(c(x')\right) \neq \mathrm{MLLM}(c(x))$

Thus, the factual shift corresponds to moving outside the semantic neighborhood while altering the atomic fact, which necessarily changes the model's reasoning-relevant conceptual representation. The only difference between the two factual shifts is whether the prompts share part of the conceptual framing, *e.g.,* the same entities, question structure, or visual context.

# D EXPERIMENTAL SETUP DETAILS

## D.1 MLLM BACKBONES

**BLIP2-OPT.** Li et al. (2023) is a vision-language pre-training framework that leverages frozen pre-trained image encoders and large language models bridged by a lightweight Querying Transformer. Our setup uses ViT-L for the vision encoder and an unsupervised-trained OPT model with 2.7 billion parameters as the decoder-based language model.

**MiniGPT-4.** Zhu et al. (2023) is a vision-language model that integrates a frozen visual encoder with the frozen Vicuna language model built on LLaMA. The model employs a single projection layer to align visual features with Vicuna and uses the same pre-trained vision component as BLIP-2, consisting of ViT-G/14 from EVA-CLIP and a Q-Former. Our setup uses ViT-G/14 for the vision encoder and a forzen Vicuna model with 7 billion parameters as the decoder-based language model.

## D.2 DATASET STRUCTURES

The reliance of `ODEdit` on three distinct data splits ($\mathcal{D}_{IN}$, $\mathcal{D}_{SE}$, $\mathcal{D}_{out}$) is not a new imposition but rather a formalization of the training datasets from benchmark MMEdit (Cheng et al., 2023), which is also the most popularly used benchmark in previous work (Pan et al., 2024). The MMEdit benchmark that we use explicitly provides data structured as triplets for each edit instance in the training datasets, i.e., the original edit sample (our $\mathcal{D}_{IN}$), semantically rephrase samples (our $\mathcal{D}_{SE}$), and unrelated samples (our $\mathcal{D}_{out}$). For clarity, here we provide the data structure of a training instance example:

---

**src:** A photo of
**pred:** Wooden spoons and forks on a wooden table.
**rephrase:** Provide a brief overview of the image content.
**alt:** A selection of wooden kitchen tools on a counter.
**image:** val2014/COCO_val2014_000000386164.jpg
**image rephrase:** val2014_image_rephrase/COCO_val2014_000000386164.png
**loc:** Who was supported by the united states during mexican civil war?
**loc ans:** Benito Juárez.
**m_loc:** val2014/COCO_val2014_000000297147.jpg
**m_loc_q:** What sport can you use this for?
**m_loc_a:** Motocross.

---

## D.3 BASELINE METHODS

To thoroughly evaluate the effectiveness of our model `ODEdit`, we compare it with four types of baselines: (1) *Naive fine-tuning*: FT directly tunes the last three layers of MLLM. (2) *Parameter-adjusting unimodal editing*: MEND (Mitchell et al., 2021). (3) *Model-extending unimodal editing*: IKE (Zheng et al., 2023), SERAC Mitchell et al. (2022), T-Patcher (Huang et al., 2023), WISE (Wang et al., 2024a). (4) *Integrate parameter-adjusting and model-extending editing*: UniKE (Pan et al., 2024). `ODEdit` serves as a plug-and-play universal framework, capable of being seamlessly integrated into any editing model that relies on loss-based optimization. Thus, we enhance one representative model under each type of baselines using `ODEdit`, *i.e.,* WISE+`ODEdit`, MEND+`ODEdit`, T-Patcher+`ODEdit`, UniKE+`ODEdit`, and compare the results against the original models.

**Fine-tune (FT).** Fine-tuning is the predominant paradigm for adapting pre-trained models to downstream tasks. As our baseline for multimodal editing, we adopt vanilla fine-tuning by updating the last three layers of the MLLM.

**In-context Knowledge Editing (IKE).** Zheng et al. (2023) explores in-context learning (ICL) for knowledge editing in large language models. IKE designs demonstration templates, *i.e.,* copy, update, retain, and retrieves relevant facts from the training corpus to construct effective in-context demonstrations that guide LLMs in precise knowledge editing.

**SERAC.** Mitchell et al. (2022) develops a memory-based editing framework, where edits are cached in an explicit memory and retrieved at inference. A scope classifier decides whether the input falls within memory coverage. When the input falls within memory coverage, it is augmented with the most relevant memory entry and forwarded to a counterfactual model for prediction.

**WISE.** Wang et al. (2024a) introduces a dual-parametric memory with a main memory for pretrained knowledge and a side memory for edits. A router determines which memory to access for each query. To support continual editing, WISE adopts sharding and merging mechanisms that isolate edits in different parameter subspaces and integrate them without conflicts.

**MEND.** Mitchell et al. (2021) designs model editor networks with gradient decomposition, a scalable approach for fast post-hoc editing of large pre-trained language models. Instead of directly fine-tuning model parameters, MEND employs lightweight auxiliary networks to transform fine-tuning gradients, using a low-rank decomposition to keep the transformation tractable. We set the last three layers of MLLM as the tuned target for this auxiliary network in our experiments.

**T-Patcher.** Huang et al. (2023) proposes a lightweight approach for model editing, aimed at revising transformer-based pre-trained language models without affecting overall performance. Instead of updating all parameters, Transformer-Patcher adds a small set of trainable neurons, *i.e.,* patches, to the FFN layer, and trains them with activation and memory losses to respond only to targeted inputs.

**UniKE.** Pan et al. (2024) presents a unified framework for multimodal knowledge editing by combining intrinsic memory updates and external memory resorting. Both types of knowledge are represented as key-value memories and edited in the latent space. Contrastive learning disentangles semantic and truthfulness aspects, allowing intrinsic and external knowledge to guide each other.

### D.4 IMPLEMENTATION DETAILS

For the generality risk in `ODEdit`, we employ a Gaussian RBF kernel with a multi-scale bandwidth strategy as the kernel function for MMD. To achieve adaptive TV-$\ell_1$ penalty, we utilize a three-layer MLP with ReLU activations for the IRM-TV optimization, Xavier initialization for weights, and Softplus activation at the output to ensure positivity. We choose Adam as the optimizer, and vary the learning rates in $\{0.0001, 0.001, 0.005, 0.01\}$ for the IRM-TV network. For all experiments, we repeat them five times and report the mean value of the results. We conduct all of our experiments on an Ubuntu OS that contains 8 NVIDIA A40 GPUs.

### D.5 PRINCIPLED GUIDELINES FOR SETTING PARAMETERS

**For TV penalty $\lambda$.** We clarify that the coefficient of the TV penalty, denoted by $\lambda$, is not a fixed hyperparameter that requires grid search. *Instead, it is treated as a Lagrange multiplier and is adaptively learned during training.* This adaptive dual-variable treatment avoids additional hyperparameter tuning for three reasons: 1) $\lambda$ is not manually chosen, and it is automatically adjusted via gradient ascent on the constraint violation. 2) The MLP parameterization for $\lambda$ is intentionally low-capacity (e.g., 2 layers, 32 units) 3) The dual update is stable across a wide range of $\gamma_2$, *i.e.,* $1\mathrm{e}^{-4}$ to $5\mathrm{e}^{-3}$, and standard techniques, *e.g.,* gradient clipping and EMA smoothing ensure robust behavior. Thus, the learned TV penalty acts as a self-regulating mechanism rather than a hand-tuned hyperparameter, and introduces negligible additional tuning workload in practical use.

**For MMD bandwidths $\sigma_q$.** In our implementation, the bandwidth used by the RBF-based MMD is computed directly from data rather than specified as a tunable hyperparameter. Given a batch of source features $\{x_i\}_{i=1}^n$ and target features $\{y_j\}_{j=1}^m$, we concatenate them into $\mathcal{Z} = \{z_k\}_{k=1}^{n+m} = \{x_1, \ldots, x_n, y_1, \ldots, y_m\}$. We then compute all pairwise squared Euclidean distances $d_{ij} = \|z_i - z_j\|^2, 1 \leq i, j \leq n + m$. The base bandwidth $\sigma^2$ is defined as the average pairwise distance (excluding diagonal entries) $\sigma^2 = \frac{1}{(n+m)(n+m-1)} \sum_{i \neq j} d_{ij}$.

This formulation makes the bandwidth fully data-adaptive, as it automatically reflects the intrinsic scale of the representations in each batch. Thus, the MMD kernel requires no manual tuning of $\sigma$ and does not introduce additional difficulty in hyperparameter optimization. To further enhance robustness, we employ a multi-scale Gaussian kernel using a geometric progression of bandwidths. Let $K$ denote the number of kernels and $\kappa$ the multiplicative factor. After normalizing the base bandwidth by $\kappa^{\lfloor K/2 \rfloor}$, we generate a set of kernel bandwidths $\sigma_k^2 = \sigma^2 \cdot \kappa^{k - \lfloor K/2 \rfloor}$, $k = 0, 1, \ldots, K - 1$. Each scale defines an RBF kernel $k_k(z_i, z_j) = \exp\left(-\frac{d_{ij}}{\sigma_k^2}\right)$, and the final kernel matrix is obtained by summing across scales $K(z_i, z_j) = \sum_{k=0}^{K-1} k_k(z_i, z_j)$. This multi-scale construction ensures sensitivity to both small and large variations in the feature representations and effectively prevents kernel collapse, i.e., a single bandwidth becomes either overly peaked or nearly constant.

**For primal-dual learning rate $\gamma_1$ $\gamma_2$ and the stablity of the primal-dual optimization.** From the experiments results, as shown in Figure 5, we observe that the primal-dual learning rates are not sensitive in practice. This is primarily due to the smoothness and boundedness of our constraint terms. The primal update optimizes a standard editing loss combined with softly-weighted regularizers, which results in well-behaved gradients. Formally, the primal step is $\phi_e^{(k+1)} = \phi_e^{(k)} - \gamma_1^{(k)} \cdot \partial_{\phi_e} \mathcal{G}(\delta^{(k)}, \phi_e^{(k)})$ and the gradient $\partial_{\phi_e} \mathcal{G}$ remains Lipschitz-continuous because both the MMD and TV terms are smooth with respect to $\phi_e$. For the dual variable, the update takes the form $\delta^{(k+1)} = \delta^{(k)} + \gamma_2^{(k)} \cdot \nabla_\delta \mathcal{G}(\delta^{(k)}, \phi_e^{(k+1)})$. The dual signal $\nabla_\delta \mathcal{G}$ reduces to the constraint violation term such as $\mathrm{TV}(\theta) - \tau$, which is naturally bounded due to gradient clipping and the compact support of the kernel function used in the MMD constraint. As a consequence, the dual gradient magnitude is inherently constrained, making the update stable over a broad range of $\gamma_2$. Empirically, we find that any $\gamma_2$ within $1\mathrm{e}^{-3}$ to $5\mathrm{e}^{-3}$ yields nearly identical behaviors: $\lambda$ grows only when the constraint is violated and quickly plateaus once the constraint is satisfied. This monotonicity property acts as an automatic stabilizer, preventing oscillation even when $\alpha_{\mathrm{d}}$ varies within a wide range. Finally, the EMA smoothing and non-negativity projection applied to $\lambda$ further dampen sensitivity. These properties ensure that both primal and dual updates behave predictably, and the optimization remains robust even when the learning rates are perturbed by one or two orders of magnitude.

**Principled guidelines for parameter setting.** Across all models and datasets we tested, we found the following configuration consistently stable and near-optimal:

- **Primal LR for editor parameters: identical to the backbone fine-tuning LR.**

- **Dual LR: $1\mathrm{e}^{-3}$ to $5\mathrm{e}^{-3}$, smaller than primal LR for stability.**

- **MMD bandwidth: median heuristic on the in-domain samples.**

- **TV penalty scale: initially set such that the initial TV magnitude is comparable to the editing loss, and then adaptively learned by MLP.**

- **MLP for TV penalty: fixed 2-layer MLP network with dim as [32,8,1].**

### D.6 INTERPRETABILITY STUDIES

To evaluate the detailed effects of Maximum Mean Discrepancy Alignment and Edit Trajectory Invariant Learning, we apply the WISE method and the WISE+ODEdit method on BLIP-2 OPT to conduct interpretability studies. Figure 4 shows several qualitative cases.

For the generality evaluation, ODEdit eliminates the spurious environmental factor, *i.e.,* window, and produces generalized answers for rephrase prompts, while editing only with baseline fails to discriminate factual shifts and loses the critical invariant feature, *i.e.,* desk. For image and text locality, ODEdit preserves accurate answers after editing, owing to the edit trajectory invariant learning. In contrast, the cognition of MLLM on irrelevant samples is affected by editing in the baseline, leading to off-topic responses.

# E  RELATED WORK

## E.1  OUT-OF-DISTRIBUTION GENERALIZATION

OOD generalization is a core challenge in machine learning, aiming for generalization under covariate shift without access to data in the target domain (Muandet et al., 2013; Arjovsky et al., 2019). The mainstream works (Arjovsky et al., 2019; Krueger et al., 2021; Ahuja et al., 2020; Lai & Wang, 2024) utilize invariant risk minimization with regularizer to explore invariant representations across different training environments. Further, a wide range of techniques is leveraged to extract and generalize invariant features (Yu et al., 2023), *e.g.,* context-based augmentation (Nam et al., 2021), representation alignment (Dou et al., 2019; Ruan et al., 2021), gradient manipulation (Shahtalebi et al., 2021), distributional robust optimization (Ghosal & Li, 2023), and meta-learning (Chen et al., 2023). In this paper, we make the first attempt to cast MLLM editing as an OOD generalization problem, where invariant learning across editing environments is enforced via a total invariance regularizer on cross-modal semantic features, so as to improve editing robustness and adaptability.

# F  LIMITATIONS AND FUTURE WORK

While ODEdit presents a robust framework for multimodal knowledge editing, our work has certain limitations that point to valuable future research directions.

**Granularity of Invariance..** Our method learns invariant trajectories at a relatively macroscopic level, *e.g.,* across semantic neighbors. The framework does not explicitly model or enforce invariance at a more fine-grained or neuron-level within the MLLM, which could be a future pathway for achieving even more precise and disentangled edits.

**MLLM Scale.** Our empirical study is confined to the multimodal large language models established in the MMEdit benchmark. Consequently, the effectiveness of ODEdit on more larger-scale, state-of-the-art MLLMs remains an open question. Extending the evaluation to more powerful and diverse architectures is a crucial direction for future work.

# G  ETHICS STATEMENT

**Ethical Impacts.** This work poses no ethical concerns, as it relies solely on publicly available datasets and models for experimentation and does not involve subjective evaluation or private data.

**Societal Impacts.** This work introduces a robust framework for editing knowledge in multimodal large language models (MLLM) from an out-of-distribution generalization perspective. The primary positive social impact of this technology is its potential to significantly enhance the reliability and safety of MLLM by enabling precise, controlled updates to their knowledge base. This is particularly critical for applications in domains such as healthcare, education, and news dissemination, where maintaining factual accuracy and mitigating harmful hallucinations are of utmost importance.

# H  REPRODUCIBILITY STATEMENT

To ensure the reproducibility of our work, we have taken the following steps. The source code for ODEdit, including implementations of the tripartite OOD risk and the Edit Trajectory Invariant Learning algorithm, has been made publicly available as anonymized supplementary material (link provided in the abstract, Our code is available at https://anonymous.4open.science/r/ODEdit-2756.). Complete theoretical proofs for our key propositions, including the equivalence between the OOD and IRM-TV objectives, are provided in Appendix A. Detailed descriptions of the experimental setup, including the MLLM backbones (Appendix B.1), baseline methods (Appendix B.2), hyperparameter configurations, and training procedures, are thoroughly documented in Appendix B.3. The MMEdit benchmark used for evaluation is publicly available, and our data processing steps are clearly outlined in Section 4.1 and Appendix B. We hope these resources will facilitate the replication and extension of our work.

# I  USE OF LLMS IN WRITING

We used a large language model (LLM) solely to polish the writing and correct grammatical issues during the preparation of this paper. The LLM was not involved in idea generation, experiment design, or analysis, and all scientific contributions are entirely made by the authors.

