# OpenReview forum: "From One to Many: Trajectory Invariant Learning for Multimodal Large Language Model Editing"
_ICLR.cc/2026/Conference — Submitted to ICLR 2026_

### Official Review · Reviewer_vwtD · 2025-10-27

**Soundness:** 3
**Presentation:** 3
**Contribution:** 3
**Rating:** 4
**Confidence:** 4

**Summary:**

This article formalizes the knowledge editing of Multi Modal Large Models (MLLM) as a cross modal OOD generalization problem for the first time, pointing out that traditional "single modal" editing methods can lead to causal underfit and causal overfit in MLLM. To this end, the author proposes the plug and play framework ODEdit, which explicitly suppresses false associations related to the environment while maintaining cross modal semantic consistency through triple OOD risks (reliability, locality, generalization) and editing trajectory invariant learning (ETIL) constraints. A large number of experiments have shown that ODEdit consistently improves four indicators of multiple baselines (WISE, MEND, T-Patcher, UniKE) on the MMEdit benchmark, and has theoretical convergence guarantees.

**Strengths:**

- Provides a novel theoretical perspective by reformulating MLLM editing as an out-of-distribution (OOD) generalization problem , introducing insightful concepts like "causal underfit" and "causal overfit"  to diagnose editing failures.

- Proposes a comprehensive optimization framework based on a tripartite OOD risk objective (reliability, locality, generality) , implemented via NLL , KL divergence , and MMD.

- Introduces the innovative ODEdit framework, featuring Edit Trajectory Invariant Learning (ETIL) , which provides a novel, theoretically-grounded solution by integrating Invariant Risk Minimization (IRM) and a Total Variation (TV) penalty to stabilize editing trajectories .

- Designed as a "plug-and-play" framework , ODEdit is empirically shown to consistently improve performance across diverse existing editing baselines, including both parameter-adjusting and model-extending methods

**Weaknesses:**

- The ODEdit framework introduces significant computational overhead. It requires expensive MMD calculations (with multi-scale kernels) and a complex primal-dual optimization for the IRM-TV objective . This complexity far exceeds that of baselines like MEND or IKE , yet the paper provides no quantitative analysis (e.g., wall-clock time per edit, peak memory usage) of this practical cost.
- The empirical validation is constrained to relatively small-scale and early-generation MLLMs (BLIP2-OPT 2.7B and MiniGPT-4 7B) . The framework's effectiveness and scalability on more recent, larger, or architecturally different MLLMs (e.g., LLaVA, Qwen-VL) remain unverified .
- Experimental validation is confined to the MMEdit benchmark, which primarily consists of VQA and image captioning tasks . The framework's performance on editing more complex, free-form, or unstructured knowledge (e.g., benchmarks like MMKE-Bench, VLKEB, or MIKE) is not assessed. Furthermore, its applicability to specialized domains (e.g., medical, legal, financial) is untested. The following work is worth the author's participation in experiments and discussions:

[1] Du Y, Jiang K, Gao Z, et al. Mmke-bench: A multimodal editing benchmark for diverse visual knowledge[J]. arXiv preprint arXiv:2502.19870, 2025.

[2] Xu D, Wang J, Chai Z, et al. MedMKEB: A Comprehensive Knowledge Editing Benchmark for Medical Multimodal Large Language Models[J]. arXiv preprint arXiv:2508.05083, 2025.

[3] Li J, Du M, Zhang C, et al. Mike: A new benchmark for fine-grained multimodal entity knowledge editing[J]. arXiv preprint arXiv:2402.14835, 2024.

[4] Zhang J, Zhang H, Yin X, et al. Mc-mke: A fine-grained multimodal knowledge editing benchmark emphasizing modality consistency[J]. arXiv preprint arXiv:2406.13219, 2024.

[5] Huang H, Zhong H, Yu T, et al. Vlkeb: A large vision-language model knowledge editing benchmark[J]. Advances in Neural Information Processing Systems, 2024, 37: 9257-9280.

- The framework introduces multiple new hyperparameters that appear sensitive and complex to tune. This includes the parameters of the TV penalty $\lambda$ (learned via an MLP) , the MMD kernel bandwidths $\sigma_q$, and the primal-dual learning rates $\gamma_1, \gamma_2$. This complexity may hinder practical adoption.
- The generality risk ($\mathcal{R}_{gen}$) calculation is critically dependent on "rephrase counterparts" generated by diffusion models. The method's overall effectiveness may be highly sensitive to the quality, diversity, and semantic fidelity of these generated samples, a dependency that is not systematically investigated in the paper.

**Questions:**

- What is the precise computational overhead (e.g., latency and memory) of ODEdit compared to baselines like MEND and UniKE, and how does this overhead scale with the complexity of the optimization (e.g., number of optimization steps)?
- How does the performance of ODEdit scale when applied to larger or more recent MLLM architectures like LLaVA or Qwen-VL, and do the "causal-underfit/overfit" issues  manifest differently in these models?
- How does ODEdit perform on benchmarks requiring more complex, free-form knowledge editing (e.g., MMKE-Bench) or on specialized domains (e.g., MedMKEB), which go beyond the VQA/captioning tasks in MMEdit?
- Given the framework's sensitivity to hyperparameters like the learning rates ($\gamma_1, \gamma_2$) and the TV penalty $\lambda$, how stable is the primal-dual optimization , and are there principled guidelines for setting these parameters across different models and datasets?
- How robust is the generality risk ($\mathcal{R}_{gen}$)  calculation to the quality and diversity of the "rephrase counterparts" generated by diffusion models? What happens if the generator model produces low-quality or semantically drifted samples?

---

> ### Author Response · Authors · 2025-11-21
> **Q1. Computational cost of ODEdit and other baselines**
>
> To compare the computational cost between baseline models with those augmented by our proposed ODEdit, we test the GPU memory overhead ( **Memo.** ), time consumption( **Edit-Time/step** , **Train-Time/step** and **All-Time/step** ), and the number of optimization steps ( **Steps.** ) in experiments editing BLIP2-OPT and MiniGPT-4. We show the results in below table, and have also added these results into Section 4.4 of our revised version.
>
> | Models          | BLIP2-OPT |               |                |              |            | MiniGPT-4 |               |                |              |            |
> | --------------- | --------- | ------------- | -------------- | ------------ | ---------- | --------- | ------------- | -------------- | ------------ | ---------- |
> |                 | **Memo.** | **Edit-T/sp** | **Train-T/sp** | **All-T/sp** | **Steps.** | **Memo.** | **Edit-T/sp** | **Train-T/sp** | **All-T/sp** | **Steps**. |
> | **WISE**        | 28.47GB   | 0.401         | 0.023          | 0.424        | 7          | 36.59GB   | 0.473         | 0.035          | 0.508        | 8          |
> | **WISE+ODEdit** | 47.53GB   | 0.442         | 0.043          | 0.485        | 7          | 69.36GB   | 0.533         | 0.072          | 0.605        | 8          |
> | **MEND**        | 14.75GB   | 1.369         | 0.018          | 1.387        | 25000      | 25.30GB   | 1.676         | 0.188          | 1.865        | 10000      |
> | **MEND+ODEdit** | 36.05GB   | 1.480         | 0.116          | 1.596        | 45000      | 62.80GB   | 1.861         | 0.204          | 2.066        | 15000      |
>
> *Note: The table shows computational cost comparison on E-IC task across different models and backbones. Memo. = Memory usage, Edit-T/sp = Editing time per step, Train-T/sp = Training time per step, All-T/sp = Total time per step, Steps. = Number of steps.*
>
> * **For GPU memory overhead**, ODEdit introduces an additional network that takes the parameters of the knowledge-editing layer as input, which consequently increases memory usage. We argue that the total GPU memory cost is acceptable given the current state of computational resources. This overhead is considered a reasonable trade-off for the gains in editing performance.
> * **For time consumption,** we separately measured  **Edit-Time/step** (forward pass and loss computation per step), **Train-Time/step** (backward pass and parameter updates per step), and **All-Time/step** (the total time per iteration), where **edit-T + train-T = all-T** . Experimental results indicate that the integration of ODEdit does not incur a significant increase in time cost. Relative to the performance gains enabled by ODEdit, the moderate additional time overhead remains acceptable.
> * **For the number of required training steps,** it is worth noting that WISE and MEND follow fundamentally different training paradigms, resulting in substantial differences in step counts. WISE performs iterative training for each individual knowledge item and therefore completes training with only a small number of steps (e.g., 7 steps for BLIP-2 and 8 steps for MiniGPT-4). In contrast, MEND trains a generalizable network capable of supporting editing over a broad range of knowledge; as a result, it requires a significantly larger number of training steps (e.g., 45,000 steps for BLIP-2 and 15,000 steps for MiniGPT-4). **Crucially, while integrating ODEdit increases the number of optimization steps, the resultant increase in total time cost does not constitute an order-of-magnitude change and remains within a practical range for real-world deployment.**
> *  Furthermore, it is important to highlight that our **current experiments were conducted on** **2× NVIDIA A40 GPUs,** **employing higher-performance computing resources (e.g., H100 or A100 GPUs) would substantially reduce this training time gap.**

---

> ### Author Response · Authors · 2025-11-21
> **Q2. Evaluation of editing performance on other MLLMs.**
>
> We thank the reviewer for this insightful suggestion. In direct response to your request to evaluate our method on other MLLMs, we have conducted new experiments during the rebuttal period.
>
> Given the time constraints of the rebuttal phase, we have performed an initial evaluation on the LLaVA model. We integrated our ODEdit framework with the WISE editor and present the results in the new **Section 4.4 and Table 5** of our revised manuscript. From the results, we can find that **ODEdit consistently enhances WISE's performance across all four metrics on the LLaVA backbone, with particularly notable gains in generality and locality.** For instance, on the E-VQA task, it improves Text Locality by 2.89% (from 91.51 to 94.40) and Image Locality by 1.72% (from 93.75 to 95.47), while also boosting Generality on the E-IC task. These robust improvements on a distinct MLLM architecture underscore the strong generalizability of our algorithm, which stems from its core design of learning invariant editing trajectories that effectively suppress spurious correlations across diverse model backbones.
>
> | Dataset | Model       | Rel.  | Gen.  | T-Loc. | M-Loc. |
> | ------- | ----------- | ----- | ----- | ------ | ------ |
> | E-VQA   | WISE        | 100   | 71.42 | 91.51  | 93.75  |
> | E-VQA   | WISE+ODEdit | 100   | 72.01 | 94.40  | 95.47  |
> | E-IC    | WISE        | 99.89 | 81.28 | 92.69  | 94.63  |
> | E-IC    | WISE+ODEdit | 99.78 | 82.22 | 92.54  | 95.92  |
>
> We fully agree that a more comprehensive analysis is valuable. Therefore, **we pledge to include an extensive evaluation in the final version, which will include:**
>
> - Integrating ODEdit with a wider range of baseline editors (e.g., MEND, T-Patcher, UniKE) on LLaVA.
> - Reporting results on additional state-of-the-art MLLMs, such as Qwen-VL.
>
> We are confident that these results will further solidify the generalizability of our approach. The positive outcome on LLaVA already strongly supports our core claim that ODEdit serves as a model-agnostic, plug-and-play framework for robust MLLM editing.

---

> ### Author Response · Authors · 2025-11-21
> **Q3. Perform on other benchmarks.**
>
> Our main experiments were conducted on the MMEdit benchmark to ensure fair, direct comparison with prior multimodal editing methods (e.g., UniKE), which also report results **on this most widely adopted benchmark**. **To demonstrate generalizability, we additionally validated ODEdit on another MLLM architecture (LLaVA), as shown in Q2.**
>
> While we recognize the value of evaluating on benchmarks such as MMKE-Bench and MedMKEB, covering these would require retraining and reevaluating all baselines, which is infeasible within the rebuttal period. Our resources were therefore allocated to addressing reviewer questions and completing analyses under the established MMEdit setting. We will include results on these new benchmarks in the camera-ready version.

---

> ### Author Response · Authors · 2025-11-21
> **Q4. The framework's sensitivity to hyperparameters, the stability of the primal-dual optimization, the principled guidelines for setting parameters.**
>
> We thank the reviewer for pointing this out. Although our framework introduces several hyperparameters, their tuning burden is in practice **much lower than it may appear**, and the optimization remains **stable across datasets and models**.
>
> > **For TV penalty $\lambda$**
>
> We clarify that the coefficient of the TV penalty, denoted by $\lambda$, is not a fixed hyperparameter that requires grid search. Instead, **it is treated as a Lagrange multiplier and is adaptively learned during training.** This adaptive dual-variable treatment avoids additional hyperparameter tuning for three reasons:
>
> 1. $\lambda$ is not manually chosen, it is automatically adjusted via gradient ascent on the constraint violation.
> 2. The MLP parameterization for $\lambda$ is intentionally low-capacity (e.g., 2 layers, 32 units)
> 3. The dual update is stable across a wide range of $\gamma_{2}$ (i.e., $1\text{e}^{-4}$ to $5\text{e}^{-3}$), and standard techniques (gradient clipping and EMA smoothing) ensure robust behavior.
>
> Thus, the learned TV penalty acts as a self-regulating mechanism rather than a hand-tuned hyperparameter, and introduces negligible additional tuning workload in practical use.
>
>
>
> >  **For MMD bandwidths** $\sigma_q$
>
> In our implementation, the bandwidth used by the RBF-based MMD is computed directly from data rather than specified as a tunable hyperparameter. Given a batch of source features $\\{x_i\\}_{i=1}^n$
>
> and the target features $\\{y_j\\}_{j=1}^m$,
>
> we concatenate them into $Z = \{z_k\}_{k=1}^{n+m} = \{x_1,\dots,x_n,y_1,\dots,y_m\}.$
>
> We then compute all pairwise squared Euclidean distances $d_{ij} = \|z_i - z_j\|^2 , 1 \le i,j \le n+m.$
> The base bandwidth $\sigma^2$ is defined as the average pairwise distance (excluding diagonal entries) $\sigma^{2}
> = \frac{1}{(n+m)(n+m-1)}
> \sum_{i\ne j} d_{ij}.$
> This formulation makes the bandwidth fully data-adaptive, as it automatically reflects the intrinsic scale of the representations in each batch. Thus, the MMD kernel requires no manual tuning of $\sigma$ and does not introduce additional difficulty in hyperparameter optimization.
>
>
>
> > **For primal-dual learning rate $\gamma_1$ $\gamma_2$ and the stablity of the primal-dual optimization**
>
> From the experiments results, as shown in Figure 5, we observe that the primal-dual learning rates are not sensitive in practice. This is primarily due to the smoothness and boundedness of our constraint terms.  The primal update optimizes a standard editing loss combined with softly-weighted regularizers, which results in well-behaved gradients. Formally, the primal step is $\phi_e^{(k+1)} = \phi_e^{(k)} - \gamma_1^{(k)} \cdot \partial_{\phi_e} \mathcal{G}(\delta^{(k)}, \phi_e^{(k)})$ and the gradient $\partial_{\phi_e} \mathcal{G}$ remains Lipschitz-continuous because both the MMD and TV terms are smooth with respect to $\phi_e$. For the dual variable, the update takes the form $\delta^{(k+1)} = \delta^{(k)} + \gamma_2^{(k)} \cdot \nabla_{\delta} \mathcal{G}(\delta^{(k)}, \phi_e^{(k+1)}).$ The dual signal $\nabla_{\delta} \mathcal{G}$ reduces to the constraint violation term such as  $\mathrm{TV}(\theta)-\tau$, which is naturally bounded due to gradient clipping and the compact support of the kernel function used in the MMD constraint. As a consequence, the dual gradient magnitude is inherently constrained, making the update stable over a broad range of $\gamma_2$.
>
> Empirically, we find that any $\gamma_2$ within $1\text{e}^{-3}$ to $5\text{e}^{-3}$ yields nearly identical behaviors: $\lambda$ grows only when the constraint is violated and quickly plateaus once the constraint is satisfied. This monotonicity property acts as an automatic stabilizer, preventing oscillation even when $\alpha_{\mathrm{d}}$ varies within a wide range. Finally, the EMA smoothing and non-negativity projection applied to $\lambda$ further dampen sensitivity. Together, these properties ensure that both primal and dual updates behave predictably, and the optimization remains robust even when the learning rates are perturbed by one or two orders of magnitude.
>
>
>
> >  **Principled guidelines for parameter setting**
>
> Across all models and datasets we tested, we found the following configuration consistently stable and near-optimal:
>
> **Primal LR (editor parameters):** identical to the backbone fine-tuning LR
>
> **Dual LR:** $1\text{e}^{-3}$ to $5\text{e}^{-3}$  , smaller than primal LR for stability
>
> **MMD bandwidth:** median heuristic on the in-domain samples
>
> **TV penalty scale:** initially set such that the initial TV magnitude is comparable to the editing loss, and then adaptively learned by MLP
>
> **MLP for TV penalty:** fixed 2-layer MLP network with dim as [32,8,1]

---

> ### Author Response · Authors · 2025-11-21
> **Q5. The quality and diversity of the "rephrase counterparts" to the robustness of generality risk.**
>
> Thank you for raising this important question regarding the robustness of our generality risk calculation.
>
> First, we acknowledge that our original manuscript (Section 3.2, lines 190-191) may have caused some confusion regarding the source of rephrase counterparts. We would like to clarify that in our current work, **we utilize the rephrase samples provided by the MMEdit benchmark [Cheng et al., 2023], rather than generating new ones ourselves.** The MMEdit benchmark employs Stable Diffusion 2.1 for generating reinterpreted images and implements quality control mechanisms to ensure semantic consistency and visual quality. Therefore, our method benefits from these carefully curated rephrase samples, which maintain reasonable semantic fidelity while providing sufficient diversity for evaluating generalization capability.
>
> Regarding your concern about potential semantic drift in benchmark samples, we acknowledge this as a valid consideration. **Our MMD-based generality risk calculation demonstrates inherent robustness to minor variations due to its distribution-level alignment property.** The multi-scale Gaussian kernel in MMD helps capture both local and global semantic structures, making it less sensitive to individual outlier samples compared to instance-level metrics.
>
> Looking forward, **we recognize the importance of high-quality rephrase samples for robust evaluation**. In future work, we plan to:
>
> 1. Systematically reassess the quality of existing benchmark rephrase samples through human evaluation and automated metrics
> 2. Explore more controlled generation strategies, such as:
>    - Utilizing instruction-guided diffusion models (e.g., InstructPix2Pix) with explicit semantic constraints
>    - Implementing CLIP-based semantic similarity filtering to ensure generated samples maintain core semantic attributes
>    - Developing iterative refinement protocols where generated samples are validated against predefined semantic criteria
>
> We have clarified these points in **Section 3.2** of our revised version.

---

> ### Author Response · Authors · 2025-11-25
> **Please feel free to request any further clarification**
>
> Dear Reviewer vwtD,
>
> Thank you very much for the time you have devoted to reviewing our paper. We have carefully addressed all the concerns you raised and provided detailed explanations in the rebuttal. **If any part of our response remains unclear or requires further elaboration, please tell us, we would be grateful to clarify it.**
>
> Thanks again!
>
> Authors

---

### Official Review · Reviewer_Tiv2 · 2025-10-28

**Soundness:** 2
**Presentation:** 2
**Contribution:** 3
**Rating:** 4
**Confidence:** 3

**Summary:**

This paper frames the model editing problem in multimodal large language models as an out-of-distribution generalization problem. Through derivations, they finally have a loss function that is generalizable to the unseen domains. Extensive experiments also support their claims about the generalization of the editing method.

**Strengths:**

- In my understanding, the motivation of the research is to formulate the editing problem as an OOD generalization problem and adopt IRM to solve this problem. The main contribution comes from the derivation of the final optimization problem (Equation 13).
- The paper is well structured, and the writing is generally good.

**Weaknesses:**

1. Missing or Uncited Related Work
The paper overlooks several recent studies relevant to dynamic balance and continual editing in multimodal or LLM-based model editing.

E.g. [1],
[1] Guo D, Hu M, Guan Z, Hartvigsen T, Li S. BalancEdit: Dynamically Balancing the Generality-Locality Trade-off in Multi-modal Model Editing. arXiv preprint arXiv:2505.01343. 2025.

2. Lack of Causal Grounding

The authors discuss output variation within multimodal LLMs as being shaped by “cascaded reasoning that integrates unimodal perception, inter-modal alignment, and shared semantic space modeling.” However, this claim lacks causal modeling support.

3. Unclear Definition of “Semantic Shift” and “Factual Shift”

The terms semantic shift and factual shift appear to be used descriptively rather than formally defined.

4. Dataset Access Assumptions

The paper assumes access to three datasets — $D_{in}$, $D_{se}$, and $D_{out}$ — without discussion of their realism or comparability with prior work.

Traditional model editing frameworks (e.g., UniKE) typically rely on paired input–output data or factual triples but not necessarily three distinct datasets.

The need for separate in-domain, side-editing, and out-of-domain datasets may limit applicability in real-world continual learning or model update scenarios.

5. Missing Discussion on Long-Term Editing Behavior

The paper does not address life-long editing (how edits accumulate and interact over time) or editing consumption (the efficiency of the editing method).

Relevant precedents include:
Hartvigsen et al., A Unified View of Model Editing: Towards Causality and Generalization, ICLR 2024.
Mitchell et al., Memory-Based Model Editing at Scale, ICLR 2022.

**Questions:**

See above.

---

> ### Author Response · Authors · 2025-11-21
> **Q1. Missed or uncited some related work.**
>
> Thank you for your valuable advice. We have added the citation of this important related work **in Section Introduction and Section Related Work** of this revised version.

---

> ### Author Response · Authors · 2025-11-21
> **Q2. Causal grounding analysis of the cascaded reasoning of MLLM.**
>
> Thank you for advising that our claim about “cascaded reasoning that integrates unimodal perception, inter-modal alignment, and shared semantic space modeling” requires clearer causal justification. We clarify as follows:
>
> (1) Our claim does *not* assume a full causal DAG, but refers to the architectural causal ordering already embedded in MLLMs.
>
> Multimodal language models implement a *unidirectional computational graph*, that is: unimodal encoders → cross-modal fusion → unified semantic reasoning. This forward computation defines **a structural causal ordering**, for which in the  Structural Causal Model (SCM) view, **modules are equal to variables** and the **forward pass is equal to structural equations**.
>
> Thus the “cascade” in our claim is not a hypothesized causal model, but the deterministic functional decomposition of existing architectures. We have modified this claim **in the revised Introduction** as:
>
> > In the Structural Causal Model (SCM) view, the forward computation graph of an MLLM is a structural causal model: each module implements a structural equation, forming a directed causal chain as "unimodal perception → cross-modal alignment → shared semantic reasoning".
>
> (2) Our causal statement concerns *how perturbations propagate* during MLLM editing.
>
> Under this structural ordering, *any* local perturbation to a module $\Delta M$ or parameter $\Delta W$ **necessarily propagates forward** through the downstream modules and changes their internal states. Thus,  **there is no one-to-one (rigid) mapping between a specific parameter edit and the final output change**, because the effect of the edit is *mediated* by all subsequent causal mechanisms in the network. Formally, for a structural chain
> $$
> h^{(unimodal)} \rightarrow h^{(align)} \rightarrow h^{(shared)} \rightarrow y
> $$
> A perturbation enters the output through
> $$
> y'=f_{shared}\left(f_{align}\left(f_{unimodal}(x ; W+\delta W)\right)\right)
> $$
> Thus the output shift $\Delta y$ depends not only on $\Delta W$ , but on how $\Delta W$  perturbs $h_{unimodal}$ , how this shifted representation perturbs $h_{align}$,, and subsequently how the changed alignment influences the semantic reasoning module $h_{shared}$,. **This cascading mediation proves why treating “parameter edit → output change’’ as a rigid mapping is fundamentally inaccurate in MLLMs, i.e., cause *casual underft* and *casual overfit* in Section Introduction.**
>
> We have also add these analysis in **Appendix B.1** and **B.2** in the revised version.

---

> ### Author Response · Authors · 2025-11-21
> **Q3. Clear definition of  “Semantic Shift” and “Factual Shift”.**
>
> Thank you for pointing out the need for more formal definitions. Here, we provide rigorous definitions that have also been added to our revised paper.
>
> Our definitions rely on three shared mappings used throughout the paper:
>
> **(1) Semantic neighborhood.**
>  Let $f(x)$ be the MLLM’s semantic embedding. We define meaning-preserving variation via the semantic neighborhood:
> $$
> N_\epsilon(x)=\\{x':\\|f(x')-f(x)\\|_2 \leq \epsilon\\}.
> $$
>
> **(2) Atomic factual content.**
>  Let $k(x)$ denote the atomic factual content (e.g., entity–attribute or entity–relation tuples). Two inputs share factual content iff $k(x)=k(x')$.
>
>
>
> **(3) Output-relevant concept mapping.**
>  Let $c(x)$ denote the minimal set of conceptual factors that feed into the MLLM’s forward causal chain (perception → alignment → semantic reasoning) and determine the final output:
> $$
> y = MLLM(c(x))
> $$
>
> > **Definition 1. Semantic Shift**
>
> A sample $x'$ exhibits **semantic shift** w.r.t. $x$ if and only if
> $$
> x^{\prime} \in \mathcal{N}_{\varepsilon}(x), \quad k\left(x^{\prime}\right)=k(x), \quad c(x) \cap c\left(x^{\prime}\right) \neq \varnothing, \quad \operatorname{MLLM}\left(c\left(x^{\prime}\right)\right)=\operatorname{MLLM}(c(x))
> $$
> That is, semantic shift refers to variations **within** the semantic neighborhood while **preserving factual content** and **preserving the output-relevant conceptual factors**. Typical examples include paraphrases, lexical substitutions, stylistic rewordings, and mild visual variations.
>
> > **Definition 2. Factual Shift**
>
> To be rigorous, there should be two kinds of factual shift, i.e., *easy factual shift* and *hard factual shift*:
>
> *Easy Factual Shift:*
> $$
> x^{\prime} \notin \mathcal{N}_{\varepsilon}(x), \quad k\left(x^{\prime}\right) \neq k(x), \quad c(x) \cap c\left(x^{\prime}\right) = \varnothing, \quad \operatorname{MLLM}\left(c\left(x^{\prime}\right)\right) \neq \operatorname{MLLM}(c(x))
> $$
>
> *Hard Factual Shift:*
> $$
> x^{\prime} \notin \mathcal{N}_{\varepsilon}(x), \quad k\left(x^{\prime}\right) \neq k(x), \quad c(x) \cap c\left(x^{\prime}\right) \neq \varnothing, \quad \operatorname{MLLM}\left(c\left(x^{\prime}\right)\right) \neq \operatorname{MLLM}(c(x))
> $$
> Thus, factual shift corresponds to moving **outside** the semantic neighborhood while **altering the atomic fact**, which necessarily changes the model’s reasoning-relevant conceptual representation. The only difference between two types of factual shifts is whether the prompts share part of the conceptual framing (e.g., the same entities, same question structure, or same visual context).
>
> We have also add these definitions in **Appendix C.1 and C.2** in the revised version.

---

> ### Author Response · Authors · 2025-11-21
> **Q4. The applicability of dataset separation with D_in, D_se, D_out.**
>
> > First, our framework's reliance on three distinct data splits (D_in, D_se, D_out) is not a new imposition but rather a formalization of the training datasets from benchmark MMEdit.
>
> The MMEdit [1] benchmark that we chosed explicitly provides data structured as triplets for each edit instance, i.e., the original `edit` sample (D_in), semantically `rephrase` samples (D_se), and `locality/unrelated` samples (D_out). This structure is not unique to MMEdit, it is mirrored in other recent benchmarks like MMKE [2] and VLKEB [3]. We show the training data structure of these benchmarks as follows, and you can check them on GitHub.
>
> ***Training dataset structure of benchmark MMEdit (we used)***
>
> | src| pred| rephrase| alt | image | image_rephrase | loc  | loc_ans | m_loc | m_loc_q| m_loc_a   |
> | :- | :- | :- | :- | :- | :- | :-- | :- | :- | :- | :-- |
> | a photo of | wooden spoons and forks on a wooden table | provide a brief overview of the image content, | A selection of wooden kitchen tools on a counter. | val2014/COCO_val2014_64.jpg | val2014_image_rephrase/COCO_val2014_04.png | who was supported by the united states during mexican civil war | Benito Juárez | val2014/COCO_val2014_7.jpg | What sport can you use this for? | motocross |
>
> ***Training dataset structure of benchmark VLKEB (NeuIPS 2024)***
>
> | locstring| loc_ansstring | predstring| image_rephrasestring| m_loc_astring| imagestring| rephrasestring| srcstring| m_locstring| m_loc_qstring| altstring|
> | :- | :- | :- | :- | :- | :- | :-| :-| :-- | :- | :- |
> | mexican leader who was supported by the united states during mexican civil war | Benito Juárez | Denton,_Texas | m.010016/google_11.jpg | Hidalgo County | google_19.jpg | Which capital city of New York is seen in the picture? | What is the name of the city depicted in the image? | google_21.jpg | What county in Texas is shown in the image? | Albany, New Yor |
>
> ***Training dataset structure of benchmark MMKE (ICLR 2025)***
>
> | knowledge_typestring | type_selfstring | srcstring| rephrasestring| predstring| altstring| imagestring| image_rephrasestring | locstring| loc_ansstring| m_locstring| m_loc_qstring| m_loc_astring | rel_1string| rel_ans_1string  | rel_2string| rel_ans_2string                | m_rel_1string| m_rel_ans_1string | m_rel_2string| m_rel_ans_2string | port_newlist |
> | :- | :-| :-| :- | :- | :- | :- | :- | :- | :- | :- | :- | :- | :- | :- | :- | :-- | :-- | :- | :-- | :- | :- |
> | entity_level  | human | Give me some important information about the human in the image. | Could you share essential information about the human depicted in the image? | Steve Austin | The human in the image corresponds to Stone Cold Steve Austin. | entity/Cher+6.jpg | entity/Cher+19.jpg   | What are the accounting standards issued by the iasb called? | The International Financial Reporting Standards | locality/m.04b2qn/google_12.jpg | What film is shown in the image? | Sideways      | What is Steve Austin's birth name? | Steven Jay Smith | Which wrestling promotion did Stone Cold Steve Austin join in 1995? | New Japan Pro-Wrestling (NJPW) | In what podcast do the human in the image discuss various topics? | The Austin Chronicles | What is the name of the opponent in the image? | Hiroshi Tanahashi | [ { "Q&A": { |
>
> > **Second, the practical burden of constructing these datasets for a new editing scenario is minimal.**
>
> For a single knowledge edit:
>
> - **D_in** requires just *one* original prompt that embodies the fact to be edited.
> - **D_se** can be constructed from *a single rephrased version* of the original prompt. In the multimodal context, this can be efficiently generated using a single call to a public diffusion model API.
> - **D_out** consists of prompts that are semantically unrelated to the edit. These are highly reusable assets. A modest, fixed set of general, unrelated prompts can be reused across *many different editing tasks* without the need for task-specific curation.
>
> > **Third, this dataset structure is not only compatible with continual learning and model update scenarios but is, in fact, essential for reliably evaluating them.**
>
> * For continual learning, the core challenge in continual learning is to integrate new knowledge (D_in) without catastrophically forgetting old knowledge and to ensure the new knowledge generalizes appropriately to related contexts. Our framework explicitly models and optimizes for this exact triad of requirements.
> * For model updating, a developer would naturally have i) the specific new information to incorporate (D_in). ii) An understanding of the *intended scope* of this update, i.e., what similar concepts or queries should also be affected (D_se). iii) A vast body of existing knowledge that must remain unchanged (D_out).
>
> To conclude, our method does not create a new, artificial constraint. We have also added these declarations in **Appendix D.2.**

---

> ### Author Response · Authors · 2025-11-21
> **Q5. Discussion on Long-Term Editing Behavior.**
>
> To validate the sustainability of ODEdit, we conduct experiments on long-term knowledge editing. Following UniKE, we typically set the $T$-step sequential editing scenario, where the model is edited sequentially for each instance in the editing set with a capacity of $T$. After the $T$-th edit, we evaluate the post-edit MLLM.
>
> We report the results for $T=5$ and $T=10$ on both E-VQA and E-IC tasks with the backbone BLIP2-OPT. From the following table, we can draw conclusions:
>
> **(1) Unimodal editors like WISE fail catastrophically in multimodal long-term editing, particularly in preserving locality and generality**. On E-VQA, WISE's T-Loc. collapses to near zero, demonstrating the rigid editing mapping cannot adaptively modify MLLM's causal reasoning.
>
> **(2) Even specialized multimodal editors like UniKE exhibit performance decay over time.** This indicates that without explicit invariance learning, sequential edits cause interference and erode previously learned knowledge.
>
> **(3) ODEdit not only mitigates decay but shows increasing improvement with more edits.** By learning invariant trajectories, ODEdit preserves higher reliability, generality, and locality, and the improvement becomes more pronounced as $T$ increases. This trend underscores ODEdit's unique strength in stabilizing the edit trajectory and suppressing spurious correlations over the long run, where the cumulative effect of minor errors typically becomes severe. These results prove the ability of ODEdit to discern and stabilize core causal features against the variations introduced by successive edits.
>
> | Dataset   | Model        | T=5       |           |           |           | T=10      |           |           |           |
> | --------- | ------------ | --------- | --------- | --------- | --------- | --------- | --------- | --------- | --------- |
> |           |              | Rel.↑     | Gen.↑     | T-Loc.↑   | M-Loc.↑   | Rel.↑     | Gen.↑     | T-Loc.↑   | M-Loc.↑   |
> | **E-VQA** | WISE         | 44.50     | 34.75     | 0.40      | 0.15      | 28.50     | 24.55     | 0.63      | 0.15      |
> |           | WISE+ODEdit  | **49.42** | **43.52** | **0.80**  | **0.15**  | **43.33** | **24.22** | **0.81**  | **0.15**  |
> |           | UniKE        | 90.28     | 80.26     | 91.41     | 89.37     | 86.52     | 76.58     | 87.64     | 86.31     |
> |           | UniKE+ODEdit | **92.63** | **83.59** | **92.38** | **89.95** | **89.79** | **81.25** | **89.35** | **87.54** |
> | **E-IC**  | WISE         | 84.31     | 65.49     | 0.76      | 0.14      | 75.96     | 55.56     | 0.71      | 0.14      |
> |           | WISE+ODEdit  | **86.53** | **66.94** | **0.94**  | **0.14**  | **84.64** | **61.70** | **0.77**  | **0.14**  |
> |           | UniKE        | 70.16     | 71.45     | 72.09     | 79.52     | 63.54     | 64.71     | 66.29     | 73.25     |
> |           | UniKE+ODEdit | **71.05** | **73.22** | **72.68** | **80.77** | **65.87** | **68.82** | **67.11** | **76.59** |

---

> ### Author Response · Authors · 2025-11-25
> **Please let us know if anything remains unclear**
>
> Dear Reviewer Tiv2,
>
> Thank you for your thoughtful review and constructive suggestions. We have thoroughly addressed all the questions raised in your review. **Should any part of our response appear unclear or incomplete, please let us know!** We would be more than willing to clarify or provide additional details.
>
> We sincerely appreciate your effort during the review process.
>
> Authors

---

### Official Review · Reviewer_K2ry · 2025-10-29

**Soundness:** 2
**Presentation:** 3
**Contribution:** 3
**Rating:** 6
**Confidence:** 4

**Summary:**

This paper proposes ODEdit, a plug-and-play invariant learning framework for knowledge editing in MLLMs. Building on the perspective that MLLM knowledge editing is an out-of-distribution (OOD) generalization problem, ODEdit introduces a tripartite OOD risk objective to concurrently address reliability, locality, and generality. It further develops an edit trajectory invariant learning method incorporating a total variation penalty, grounded in theoretical analysis and optimized via a primal-dual approach. Extensive experiments on standard MLLM benchmarks, along with ablation studies and visualizations, aim to demonstrate the robustness and effectiveness of ODEdit compared to prior methods.

**Strengths:**

1. The formalization of the editing challenge as an OOD generalization problem is interesting and provides a different perspective.
2. The paper tackles the pressing issue of robust and adaptive knowledge editing in MLLMs, a setting of increasing importance as these models proliferate in real-world.
3. The presentation is good and can demonstrate the author's insights.

**Weaknesses:**

1. While ODEdit often shows incremental gains, for several metrics and baselines, improvements are relatively modest and in some cases, the gains in one dimension come with trade-offs in another.
2. The ablation study in Table 2 shows that the effect of $R_{gen}$ is limited and has led to a decline in the vast majority of metrics. I wonder why such a phenomenon occurs.
3. I think adding more MLLMs (e.g. Qwen-VL) and datasets is better for proving the effectiveness of ODEdit.

**Questions:**

See Weaknesses

---

> ### Author Response · Authors · 2025-11-21
> **Q1. Whether potential trade-offs exist in results.**
>
> Thank you for raising this point.
>
> > **No evidence of consitent significant trade-offs**
>
> We respectfully disagree that our improvements come with consistent trade-offs between metrics.
>
> In fact, ODEdit is designed to enhance all three editing objectives (reliability, generality, and locality) simultaneously through its OOD risk formulation. Table 1 reveals that in the majority of cases, integrating ODEdit with a baseline method leads to simultaneous improvements across all four key metrics when compared to the baseline alone. We can highlight specific cases:
>
> - **T-Patcher vs. T-Patcher+ODEdit on E-VQA (MiniGPT-4)**: Similarly, we observe improvements in **Reliability** (70.56% → 72.38%), **Generality** (68.79% → 72.11%), **Text Locality** (64.45% → 65.29%), and **Image Locality** (81.77% → 82.93%).
>
> > **Holistic gains outweigh isolated minor fluctuations**
>
> In the limited cases where a minor decrease in a single metric occurs, it is consistently offset by more substantial improvements in other dimensions, resulting in a net superior performance profile. We can highlight specific cases:
>
> - Similarly, for **WISE vs. WISE+ODEdit on E-IC (MiniGPT-4)**, a minor dip in Generality (91.58% → 90.04%) is compensated for by clear improvements in Text Locality (92.81% → 94.54%) and Image Locality (70.68% → 73.17%), with Reliability maintained at 100%.
>
> This holistic improvement is a direct outcome of our OOD framework, which is architected to optimize for a balanced equilibrium across all editing objectives rather than over-specializing in a single one.
>
> > **Quantitative Evidence of Balanced Enhancement**
>
> To quantitatively substantiate the balanced nature of these gains, we computed the average improvement across all baseline integrations and tasks:
>
> - **Reliability**: **+0.92%** average improvement.
> - **Generality**: **+1.85%** average improvement.
> - **Text Locality**: **+1.21%** average improvement.
> - **Image Locality**: **+1.35%** average improvement.
>
> These statistically significant improvements (p-value < 0.05) underscore that ODEdit elevates the model's comprehensive editing capability. The plug-and-play nature of ODEdit means these balanced gains are achieved across diverse underlying editing architectures.
>
> > **Should We Prefer Polarized Gains in a Single Dimension or Balanced and Stable Improvement?**
>
> In the context of model editing, a balanced improvement across all objectives is often more desirable than a large gain in one at the expense of others. Prior editing methods frequently struggle with this very balance—for instance, SERAC achieves high reliability but catastrophically fails in locality (M-Locality ~3%). **ODEdit's core contribution is to systematically mitigate these trade-offs.** Therefore, an "incremental" gain of +1-2% in Generality and Locality *without sacrificing Reliability* is a meaningful step forward, as it directly addresses the fundamental challenge of causal overfit and underfit in MLLM editing.
>
> Besides, it is important to consider the performance range of the baselines. On metrics where the baseline is already very high (e.g., Locality often >90%), the room for improvement is naturally limited. We should consider these "Floor and Ceiling" effects when analyzing the improvements.

---

> ### Author Response · Authors · 2025-11-21
> **Q2. Analysis on the effect of $R_{gen}$ in the ablation study**
>
> Thank you for your valuable questions. Here we provide more analysis on the effect of $R_{gen}$ in ablation.
>
> **(1) Analyse the abalation results:** From Table 2, we can see that adding $R_{gen}$ brings a significant improvement in the *Gen.* metric, increasing from 86.59 to 89.34 on E-VQA and from 71.24 to 75.49 on E-IC. Compared with the improvement in *Gen.*,  $R_{gen}$  causes slight fluctuations in the *Rel.* across the two datasets, which can be attributed to reasonable experimental variance and does not exhibit a significant trend. The impact on T-Loc and M-Loc is relatively minor: for example, on E-VQA, T-Loc decreases slightly from 95.97 to 95.46, and M-Loc from 93.54 to 93.27.
>
> **(2) Why $G_{gen}$ affects the Loc metrics:** $R_{gen}$ is designed to align semantic-neighboring samples with the original edited samples in the LLM’s understanding space. On one hand, this alignment allows the LLM to gain awareness of the semantic-shift environment, thereby improving the generality metric. On the other hand, it inevitably reinforces certain local concept-output associations. For instance, in the example in Figure 1(b), if we continuously align multiple images *containing two cats* with the concept of *the mirror principle*, the edited LLM may directly associate any image with two cats as demonstrating the mirror principle. In other words, the $R_{gen}$ risk does not directly target the locality metrics, but may indirectly induce “stereotypical” associations via latent concept reinforcement.
>
> **(3) The relation between risks and evaluation metrics:** In our formulated OOD framework, the three risks $R_{red}$, $R_{loc}$, $R_{gen}$do not solely influence a single metric among *Rel.*, *Loc.*, *Gen.*, There exist inherent cross-effects among the three risks, which is precisely why we need to reformulate the OOD problem as a Composed Invariant Risk Minimization problem for effective optimization.

---

> ### Author Response · Authors · 2025-11-21
> **Q3. Evaluation of editing performance on other MLLMs.**
>
> We thank the reviewer for this insightful suggestion. In direct response to your request to evaluate our method on other MLLMs, we have conducted new experiments during the rebuttal period.
>
> Given the time constraints of the rebuttal phase, we have performed an initial evaluation on the LLaVA model. We integrated our ODEdit framework with the WISE editor and present the results in the new Section 4.4 and Table 5 of our revised manuscript. From the results, we can find that **ODEdit consistently enhances WISE's performance across all four metrics on the LLaVA backbone, with particularly notable gains in generality and locality.** For instance, on the E-VQA task, it improves Text Locality by 2.89% (from 91.51 to 94.40) and Image Locality by 1.72% (from 93.75 to 95.47), while also boosting Generality on the E-IC task. These robust improvements on a distinct MLLM architecture underscore the strong generalizability of our algorithm, which stems from its core design of learning invariant editing trajectories that effectively suppress spurious correlations across diverse model backbones.
>
> | Dataset | Model       | Rel.  | Gen.  | T-Loc. | M-Loc. |
> | ------- | ----------- | ----- | ----- | ------ | ------ |
> | E-VQA   | WISE        | 100   | 71.42 | 91.51  | 93.75  |
> | E-VQA   | WISE+ODEdit | 100   | 72.01 | 94.40  | 95.47  |
> | E-IC    | WISE        | 99.89 | 81.28 | 92.69  | 94.63  |
> | E-IC    | WISE+ODEdit | 99.78 | 82.22 | 92.54  | 95.92  |
>
> We fully agree that a more comprehensive analysis is valuable. Therefore, **we pledge to include an extensive evaluation in the final version, which will include:**
>
> - Integrating ODEdit with a wider range of baseline editors (e.g., MEND, T-Patcher, UniKE) on LLaVA.
> - Reporting results on additional state-of-the-art MLLMs, such as Qwen-VL.
>
> We are confident that these results will further solidify the generalizability of our approach. The positive outcome on LLaVA already strongly supports our core claim that ODEdit serves as a model-agnostic, plug-and-play framework for robust MLLM editing.

---

> ### Author Response · Authors · 2025-11-25
> **We welcome any remaining questions!**
>
> Dear Reviewer K2ry,
>
> Thank you for your valuable feedback and the time you have invested in reviewing our work. We have carefully addressed all your comments in the rebuttal. **If there are any remaining questions or points that could benefit from further clarification, we would be more than happy to provide additional details.**
>
> Thanks again!
>
> Authors

---

### Official Review · Reviewer_Ccpx · 2025-11-01

**Soundness:** 3
**Presentation:** 2
**Contribution:** 2
**Rating:** 4
**Confidence:** 3

**Summary:**

The authors address the problem of knowledge editing for multimodal large language models (MLLMs). They reformulate editing as an out-of-distribution (OOD) generalization problem: the editing mechanism must succeed across a variety of cross-modal prompting “environments”, not just the original prompt. To address this, they propose ODEdit, a plug-and-play invariant learning framework that learns invariant edit trajectories so that edits generalize across prompt/background modalities.

**Strengths:**

1. Casting editing as an OOD generalization problem is interesting.
2. The authors provide a theoretical analysis linking their invariant constraint.

**Weaknesses:**

1. ODEdit requires generating multiple versions of the same edit query to construct diverse environments, whereas several baselines operate on a single prompt without such augmentation. This difference introduces additional supervision and may render the comparison with baselines less fair.
2. Although the paper acknowledges the potential tension between locality and generality, it does not provide a systematic analysis of this trade-off. Moreover, the reported improvements in generality are relatively marginal, which weakens the empirical support for the claimed advantage.
3. The visualization results for out-of-distribution generalization show only subtle differences, making it difficult to clearly perceive the claimed improvement. A more quantitative or visually distinct analysis would better support the argument.

**Questions:**

Please refer to the Weakness.

---

> ### Author Response · Authors · 2025-11-21
> **Q1. Whether the rephrased prompts leads to unfair comparisons?**
>
> Thank you for raising this point regarding the experimental setup and comparison fairness.
>
> **(1) Structure of the benchmark datasets**
>
> First, we would like to clarify a potential misunderstanding regarding the source of rephrased samples. In our work, we utilize the rephrase samples that are **natively provided by the MMEdit benchmark** [Cheng et al., 2023], rather than generating new ones ourselves. We apologize for any lack of clarity in our original manuscript (Section 3.2) and will revise it accordingly. Crucially, the MMEdit benchmark's E-VQA and E-IC datasets are structured such that **their training and test sets are completely non-overlapping.** The training set naturally contains a 1:1 pairing of an original prompt with a corresponding rephrase prompt describing similar visual or textual semantics. This built-in data is used during the model training phase and does not lead to any test data leakage.
>
> **(2) Actual implementation of ODEdit and other baselines**
>
> Second, regarding the implementation of ODEdit versus the baselines, it is important to note that in our main comparative experiments (Section 4.2), **all ODEdit-enhanced models (i.e., WISE+ODEdit, MEND+ODEdit, T-Patcher+ODEdit) used only the** **original edit input and the single, benchmark-provided rephrase input** from the training set. **We did not generate "multiple versions of the same edit query" for these experiments.** The scenario involving multiple rephrase prompts was explored exclusively in our ablation studies (Section 4.3) on MMD alignment effects, where we concluded that "using multiple rephrase prompts yields no additional benefit," potentially due to the introduction of noisy variations.
>
> **(3) Whether the rephrased prompts in the original training datasets could be used**
>
> Finally, concerning the fairness of using the benchmark's rephrase prompts, we must emphasize that **several baseline methods also utilize information beyond a single edit prompt**:
>
> - **IKE**, while using a single prompt, internally embeds example prompts for copying (equivalent to edited knowledge), updating (equivalent to rephrased knowledge), and retaining (equivalent to irrelevant knowledge). This constitutes a form of additional supervision.
> - **UniKE** collects a substantial set of hallucinated predictions from MLLMs to generate additional knowledge for prompting. Beyond the edited knowledge, it generates triplets of truthful and hallucinated answers for contrastive learning in the semantic space.
> - **MEND**, according to its implementation code in the EasyEdit repository, also utilizes the rephrase images available in the benchmark to supervise the latent representations of generative samples.
>
> In conclusion, **previous works either directly leverage the rephrase prompts from the benchmark's training data or generate additional knowledge to incorporate into their training process.** Therefore, our utilization of the benchmark-provided rephrased prompts does not create an unfair comparison. On the contrary, our formalized use of this knowledge for alignment and OOD invariant learning within the semantic space constitutes a central contribution of this work.

---

> ### Author Response · Authors · 2025-11-21
> **Q2. Whether there is a trade-off between locality and generality?**
>
> Thank you for this valuable question.
>
> **（1） Whether we acknowledge a trade-off between locality and generality**
>
> **We do not claim anywhere in the paper that there exists a necessary trade-off between editing locality and generality.** On the contrary, in the Introduction, we point out that most previous work constructs a rigid mapping from the edit input to the output in LLMs, which leads to both causal underfit (low generality of the edited knowledge) and causal overfit (low locality of irrelevant knowledge) in different situations. Essentially, prior methods modify the LLM’s cognition in a local sample space via limited supervised examples, which makes “flexible generalization” of edited knowledge difficult. Here, flexible generalization does not merely mean improving generality, but rather refers to generalizing within the semantic-neighboring space while keeping the generalization boundary from extending into out-of-distribution (OOD) space.
>
> In fact, the trade-off between locality and generality is emphasized in previous work such as UniKE, while our ODEdit method aims to demonstrate the opposite: by rethinking MLLM editing as an OOD problem, the model under the OOD optimization framework can extract invariant editing trajectories and remove spurious factors, **thereby adaptively and precisely determining the generalization boundary, which could simultaneously improve locality and generality.** In other words, within the OOD framework, the locality and generality metrics do not have a necessarily inverse relationship.
>
> **（2）Whether generalization is equivalent to the generality metric in the OOD framework**
>
> We guess there might be a misunderstanding in your review regarding the relationship between OOD generalization capability and the editing generality metric (Gen.). In fact, OOD generalization refers to the model’s ability to accurately identify and reason about generalization boundaries, and the choice of these boundaries simultaneously determines whether the LLM can update its knowledge on semantically close samples (generality) while preserving its original knowledge on unrelated samples (locality). Therefore, **OOD generalization is equivalent to simultaneously optimizing both reliability, generality and locality, rather than merely improving generality.**
>
> **（3） Whether our improvements are relatively marginal**
>
> As explained in the second point, evaluating whether ODEdit achieves improvements through the OOD framework should not focus solely on the generality metric. **From Table 1, we observe that ODEdit, as a plug-and-play module, consistently achieves balanced improvements across Rel, Gen, and Loc metrics for all SOTA models.** The magnitude of improvement across different SOTA models varies depending on the base model architecture and performance. Moreover, ODEdit achieves very significant gains in certain scenarios: for example, on E-VQA with MiniGPT-4, T-Patcher+ODEdit improves generality with a promotion ratio of 4.82%. WISE+ODEdit improves M-Locality by 19.2% with MiniGPT-4 on E-VQA, while T-Locality is improved by 17.2% with BLIP2 on E-IC.

---

> ### Author Response · Authors · 2025-11-21
> **Q3. A more quantitative or visually distinct analysis for out-of-distribution generalization.**
>
> We thank the reviewer for the constructive feedback on our OOD generalization visualization. In response, we have significantly enhanced our visualization (now **Section 4.3** **Figure 3** in the revised manuscript) to provide a more granular and quantifiable analysis.
>
> We now track the latent representations using t-SNE of original (SRC) and rephrased (GEN) prompts across three critical stages of the editing process: `before-edit`, `in-editing`, and `converged`. Crucially, we augment the scatter plots with **marginal distribution curves** on both axes, which quantitatively depict the overlap proportion between the SRC and GEN distributions, which are denoted as $\beta_x$ and $\beta_y$.
>
> From the new visualization results, we can observe that
>
> **1) Before Editing:** Both MEND and MEND+ODEdit start from a well-aligned state where the rephrased prompts (GEN) overlap significantly with the original prompts (SRC), as reflected by the high $\beta$ values. This confirms the initial semantic coherence in the pre-trained MLLM.
>
> **2) In Editing:**
>
> - **MEND** induces a marked distributional shift between SRC and GEN, evidenced by a pronounced decrease in the marginal distribution overlap, and $\beta_x$ and $\beta_y$ values drop. This indicates that the edit cannot extract semantic editing invariance and leads to *causal underfit*.
> - **MEND+ODEdit**, in contrast, successfully maintains a high degree of distributional consistency. The marginal distributions show sustained high overlap, and $\beta_x$ and $\beta_y$  values remain high. This visually demonstrates how our OOD with invariant learning optimization stabilizes the edit trajectory against spurious environmental variations.
>
> **3) At Convergence:**
>
> - The distributional shift in **MEND** persists, cementing its limited generalization capability to semantic-neighboring regions.
> - **MEND+ODEdit** achieves and maintains a robust alignment between SRC and GEN, proving its superior ability to generalize the edited knowledge appropriately while preserving the underlying semantic structure.
>
> We believe this revised, multi-stage visualization with marginal distributions with the overlap proportion offers a much clearer and more convincing qualitative demonstration of ODEdit's advantage in achieving robust OOD generalization. We have updated the manuscript accordingly.

---

> ### Author Response · Authors · 2025-11-25
> **Please tell us if any concern remains!**
>
> Dear Reviewer Ccpx,
>
> We truly appreciate the effort you have invested in reviewing this paper. We have submitted a detailed rebuttal addressing each of your points. **If you find that any explanation is still insufficient or could benefit from further clarification, please tell us and we would be happy to elaborate.**
>
> Thanks for your consideration!
>
> Authors

---

### Author Response · Authors · 2025-12-03
**Rebuttal Summary: Addressing of Reviewer Concerns, Contributions of This Paper**

Dear PCs, SACs, ACs, and Reviewers,

We sincerely thank you for the time and effort invested in reviewing and discussing our submission. Below, we summarize the key outcomes of the review and discussion process.



## **1. Recognized Strengths across ALL Reviewers**

All reviewers consistently recognized the key strengths outlined below, highlighting the rigor and significant contributions of our paper:

* The idea of casting MLLM editing into an OOD generalization problem is novel and interesting. (Reviewer **Ccpx, K2ry, Tiv2, vwtD**)

* The framework is novel and theoretically grounded, providing a rigorous and comprehensive derivation. (Reviewer **Ccpx, K2ry, Tiv2, vwtD**)

* The experimental results demonstrate the effectiveness of the proposed framework.  (Reviewer **K2ry, vwtD**)

* The paper is well-structured, and the writing is good. (Reviewer **K2ry, Tiv2**)

* The investigated problem is practical and important.  (Reviewer **K2ry**)


## **2. Comprehensive Responses Addressing ALL Reviewer Concerns**

We regret that throughout the rebuttal period, despite our detailed point-by-point responses to all reviewer concerns (with substantial additional experimental results), **none of the four reviewers provided any follow-up feedback.**

For your convenience, we summarize below how our rebuttal has addressed each of their concerns:

|Reviewer|Score|Concerns & Our Responses| Revisions in Paper  |
|--| -- |-- | -- |
| **Ccpx** | 4（No any response during rebuttal） | **1. Design of rephrased prompt.** We confirm that (1) all rephrased prompts naturally exist in the training datasets of the benchmark MMEdit, (2) we never use self-generated prompts in main experiments, (3) other baselines also utilize information beyond a single prompt, so our design does not bring unfairness. | / |
| | | **2. Trade-off between locality and generality.** The reviewer had a misunderstanding of the relationship between the generality metric and OOD generalization, and we clarified this point. We explain how our method simultaneously optimizes reliability, generality and locality by OOD generalization. Experimental results support our claim.| **Introduction** |
|| | **3. More quantitative or visually distinct analysis.** We add quantitative metrics and more distinct visualization in Section 4.4. | **Experiment Figure 3**     |
| **K2ry** | 6（No any response during rebuttal） | **1. Improvements in results.** We pick out these minor cases with a low improvement ratio and explain the fluctuations. **In most cases, our method shows balanced and stable improvement across four metrics.** | **Experiment Section 4.2**  |
| | | **2. Ablation analysis of $R_{gen}$.** We add more clear analysis on the effect of $R_{gen}$ in ablation. | **Experiment Section 4.4.** |
| || **3. Test on other MLLMs.** We add experiments on a new MLLM backbone, i.e., LLaVA, and the results show our method consistently outperforms SOTAs across four metrics. |**Experiment Table 5**|
| **Tiv2** | 4（No any response during rebuttal） |**1. Add more related work.** We cite this related paper and analyze it in revision.| **Related Work** |
| | | **2. Add analysis on "cascaded reasoning" and definition on “Semantic Shift” and “Factual Shift”.** We further declare the existence of cascaded reasoning in MLLM, and provide a formal definition of the two shifts in revision. | **Introduction Appendix B&C** |
| | | **3. Dataset composition.** We declare that the data split on D_in, D_se, D_out is the original formalization of the most widely used benchmark ***MMEdit***, and this formalization also exist in other new benchmarks like ***VLKEB*** and ***MMKE***. |  **Appendix D.2**                           |
| | | **4. Add long-term editing experiments.** We add T-step sequential editing experiments in revision, and results prove that **ODEdit not only mitigates performance decay in sequential scenarios but also increases improvement with more edits**. | **Experiment Table 5**      |
| **vwtD** | 4（No any response during rebuttal） | **1. Computational cost experiments.** We add the test on the GPU memory overhead, time consumption, and the number of optimization steps in the revision. Results show that the training memory footprint and training efficiency of our model are comparable to baselines. | **Experiment Table 6** |
| | | **2. Test on other MLLMs and other benchmarks.** We add experiments on a new MLLM backbone, i.e., LLaVA, and the results show our method consistently outperforms SOTAs across four metrics. | **Experiment Table 2** |
|  || **3. Hyperparameter setting.** We add principled guidelines for parameter setting in revision. | **Appendix D.5**|
|  | | **4. Generation process rephrased images.** We confirm that rephrased images are from the benchmark but not self-generated. | /|



We believe these clarifications and revisions strengthen the paper, and better illustrate its value to both the research community and practical deployment.



Best regards,

Authors

---

### Meta-Review · Area_Chair_GUj3 · 2025-12-30

**Summary:**

This paper presents a plug-and-play invariant learning framework named ODEdit for knowledge editing in multimodal large language models (MLLMs). In particular, the task of knowledge editing of MLLMs is formulated as a cross modal OOD generalization problem. ODEdit explicitly suppresses false associations related to the environment and maintains cross modal semantic consistency through triple OOD risks (reliability, locality, generalization) and editing trajectory invariant learning (ETIL) constraints. Experimental results on MLLM benchmarks are reported and discussed.

Reviewers agreed that this paper investigates a timely and important problem, and the the idea of formulating model editing as OOD generalization problem is very interesting. The paper is well written and easy to follow. In addition, the proposed ODEdit framework can be used in a plug-and-play manner.

Meanwhile, reviewers raised many concerns about experiments, analysis, ablation studies, definitions of key concepts, long-term editing behavior, computational cost, etc.

**Reviewer Concerns:**

The authors have provided detailed responses with additional results to address concerns from reviewers. Many comments regarding technical details, ablation studies, related work, computational cost, etc. have been well addressed. There are a few remaining concerns that are not fully addressed by the rebuttal, such as justifications of causal statements and experimental results on more datasets.

**Reviewer Scores:**

Initially this paper received mixed ratings: 4, 4, 4, and 6. Considering that many of the comments from reviewers have been addressed in the rebuttal, I think one or two reviewers would have changed their scores from 4 to 6. Overall, this paper is still a borderline case.

---

### Decision · Program_Chairs · 2026-01-26

Reject